# Spin-chain based quantum thermal machines

Edoardo Maria Centamori,[1] Michele Campisi,[2] and Vittorio Giovannetti[2]

[1]*Scuola Normale Superiore, I-56126 Pisa, Italy*
[2]*NEST, Scuola Normale Superiore and Istituto Nanoscienze-CNR, I-56126 Pisa, Italy*

We study the performance of quantum thermal machines in which the working fluid of the model is represented by a many-body quantum system that is periodically connected with external baths via local couplings. A formal characterization of the limit cycles of the set-up is presented in terms of the mixing properties of the quantum channel that describes the evolution of the fluid over a thermodynamic cycle. For the special case in which the system is a collection of spin 1/2 particles coupled via magnetization preserving Hamiltonians, a full characterization of the possible operational regimes (i.e., thermal engine, refrigerator, heater and thermal accelerator) is provided: in this context we show in fact that the different regimes only depend upon a limited number of parameters (essentially the ratios of the energy gaps associated with the local Hamiltonians of the parts of the network which are in direct thermal contact with the baths).

## I. INTRODUCTION

In the last decade, enormous efforts have been devoted to investigate the manipulation of heat and work in quantum devices (see e.g. Refs. [1–6] and references therein). On one side this is motivated by the need to better understand how the energy dynamics of controlled quantum systems influences the possibility of using them to implement quantum computation and quantum information tasks. On the other side, the study of quantum thermal machines is also motivated by the expectation that quantum technologies may also have a key role in improving energetic optimization processes [7].

At the experimental level implementations of thermodynamic cycles involving few-level quantum systems have been proposed in atoms and ions set-ups [8–11], NV centers [12, 13], quantum dots (see e.g. [14, 15]), superconducting nanostructures [16–19], nano electro-and optomechanical devices [20]. Most of these setups are based on configurations where a few-level coherent quantum system, partially under control through a series of classical knobs that allow for the tuning of its Hamiltonian, plays the role of a working fluid that operates in out of equilibrium conditions due to contacts with macroscopic thermal reservoirs. At theoretical level the simplest model of this type assumes the presence of two thermal baths (one hot and one cold) which exchange heat via the dynamical evolution of the fluid. Despite its simplicity these models can be used to implement a series of fundamentally different thermal devices, specifically heat engines, refrigerators, thermal accelerator, and heaters, depending on the relative sign of of the heats and work the device exchanges with the baths and work sources, respectively [21–23]. The optimal performances of these setups has been discussed within operational assumptions, ranging from low-dissipation and slow driving regimes [24–27], to shortcuts to adiabaticity approaches [28–30] , to endoreversible engines [31].

References [32–34] studied the case where the driven working fluid is a spin chain, and focussed on the possible impact of phase transitions and long-range forces on the engine performance. In the present paper we focus on a special type of models where the working fluid is represented by a many-body quantum system that is driven by a periodic sequence of operations (strokes) that either put it in thermal contact with one of the bath, or let it freely evolve under the action of its (local) Hamiltonian, that is, at variance with previous studies, e.g. [32–34], here the working fluid is driven by an external force that is responsible for tuning on and off the local couplings. Besides that, no other external force is applied onto the system. In this setting, exploiting the quantum channel formalism [35], we show the emergence of limit cycles that force the system into out-of-equilibrium, metastable states characterized by oscillatory behaviours which depend on the details of the system Hamiltonain. For the special case in which the working fluid is represented by a network of coupled 1/2-spins particles interacting through couplings that preserve the total magnetization along the (longitudinal) $\hat{z}$ direction, we show that the analysis simplifies thanks to the inner symmetry of the model. In particular we prove that for those systems the signs of the heat fluxes of the models only depend on a limited number of the parameters (the local energies gaps associated with the part of the networks which are connected with the baths) allowing us to unveil an universal law which analytically singles out the thermodynamic character of the cycle (i.e., which type of thermal device it realizes). Furthermore, supported by a series of compelling numerical and theoretical evidences, we propose an ansatz according to which for these models the fundamental thermodynamic quantities that characterize the limit cycle can be exactly determined by just studying the low temperature response of the system. We stress that if proven correct, such conjecture will allow for a massive simplification of the problem, paving the way for the numerical treatment of models with enormous numbers of spin elements.

The manuscript is organized as follows. In Sec. II the model is introduced, together with the notions of operation

regimes and of limit cycles. In Sec. III we focus on the special case of thermal engines where the working fluid is represented by a linear chain of spin 1/2 particles, coupled via magnetization preserving Hamiltonians. Here in Sec. III A we provide the universal law that determines the thermodynamical character of the limit cycle, while in Sec. III B we present our ansatz solution that, if proven correct, will permit to determine the fundamental quantities of the limit cycle by only studying the low temperature response of the model. To enlighten the peculiarity of these models, an example of spin-chain model with Hamiltonian $H$ that does not preserve the longitudinal magnetization is finally analyzed in Sec. IV. Conclusions are given in Sec. V. The paper also includes an extended technical Appendix.

## II.    THE MODEL

We focus on quantum thermal machines whose (quantum) working fluid is represented by a tripartite quantum system $ACB$ composed by two external elements ($A$ and $B$) and by an internal one ($C$) that are coupled through a first-neighbour interaction Hamiltonian of the form

$$H := H_A + H_{AC} + H_C + H_{CB} + H_B \ , \tag{1}$$

with the $H_X$'s representing local contributions of the subsystem $X$ and with $H_{XX'}$ representing instead the coupling terms. For such model we consider thermodynamic cycles schematically represented in Fig. 1, where at regular time intervals, the $A$ and $B$ elements are put in thermal contact with external reservoirs according to the following four-stroke procedures:

1. **Thermalization with the hot bath:** Site $A$ is detached from the rest of the chain through a sudden quench and weakly coupled to a hot bath (H) at temperature $T_1$ until it reaches complete thermalization on a timescale that is negligible with respect to the free-evolution dynamics of the system (this last hypothesis is not fundamental: it is only introduced to allow for some simplification in the analysis). Indicating with $\rho_{ACB}$ the input state of the chain before the stroke, its associated output will be hence evolved through the following mapping

$$\begin{cases} \rho_{ACB} \rightarrow \mathcal{T}_1(\rho_{ACB}) := \rho_A(\beta_1) \otimes \rho_{CB}, \\[2mm] \rho_A(\beta_1) := \frac{e^{-\beta_1 H_A}}{Z_A(\beta_1)}, \qquad \rho_{CB} := \mathrm{Tr}_A[\rho_{ACB}] \ , \end{cases} \tag{2}$$

   where $\mathrm{Tr}_A[...]$ indicates the partial trace over $A$, and where $Z_A(\beta_1) := \mathrm{Tr}\big[e^{-\beta_1 H_A}\big]$ denotes the partition function of system $A$ at inverse temperature $\beta_1 := 1/(k_B T_1)$ ($k_B$ being the Boltzmann constant).

2. **Unitary evolution:** The site $A$ is attached back to the chain through a second quench, and the entire chain is left free to evolve for a time $\tau_1$ inducing the mapping

$$\begin{cases} \rho_A(\beta_1) \otimes \rho_{CB} \rightarrow \tilde{\rho}_{ACB} = \mathcal{U}_1(\rho_A(\beta_1) \otimes \rho_{CB}) := U(\tau_1)\,(\rho_A(\beta_1) \otimes \rho_{CB})\,U^\dagger(\tau_1) \\[2mm] U(\tau_1) := e^{-iH\tau_1}, \end{cases} \tag{3}$$

   (hereafter we set $\hbar = 1$).

3. **Thermalization with the cold bath:** The site $B$, detached from the rest of the chain, is put in weak coupling with a cold bath (C) of temperature $T_2 \leq T_1$ until complete thermalization is reached

$$\begin{cases} \tilde{\rho}_{ACB} \rightarrow \mathcal{T}_2(\tilde{\rho}_{ACB}) := \tilde{\rho}_{AC} \otimes \rho_B(\beta_2), \\[2mm] \tilde{\rho}_{AC} := \mathrm{Tr}_B[\tilde{\rho}_{ACB}], \qquad \rho_B(\beta_2) := \frac{e^{-\beta_2 H_B}}{Z_B(\beta_2)}, \end{cases} \tag{4}$$

   where again $Z_B(\beta_2) := \mathrm{Tr}\big[e^{-\beta_2 H_B}\big]$ and $\beta_2 := 1/(k_B T_2)$, and where once more we assume the thermalization time to be negligible.

4. **Unitary evolution:** The site $B$ is attached back to the chain, and the entire chain is left free to evolve for a time $\tau_2$

$$\begin{cases} \tilde{\rho}_{AC} \otimes \rho_B(\beta_2) \rightarrow \mathcal{U}_2(\tilde{\rho}_{AC} \otimes \rho_B(\beta_2)) := U(\tau_2)\,(\tilde{\rho}_{AC} \otimes \rho_B(\beta_2))\,U^\dagger(\tau_2), \\[2mm] U(\tau_2) := e^{-iH\tau_2}. \end{cases} \tag{5}$$

FIG. 1: Schematic representation of the typical four-strokes thermodynamic cycle of our model where we identify the system with a spin-chain of $N = 5$ qubits ($A$ and $B$ corresponding to the first and last elements). Notice that the system undergoes through a total of 4 quenches (located at the beginning of each stroke) and two thermalization events (associated with stroke 1 and stroke 3).

During this cycle, the system changes its internal energy due to energy exchanges with the reservoirs (in the form of heat) and to the quenches (in the form of work). Note that thanks to the assumption of weak-coupling with the external reservoir, any work extraction or injection related to the quenches relative to connecting or disconnecting the baths can be neglected, hence only the quenches that connect/disconnect a site to the rest of the chain are associated to some work. To illustrate this point, consider the total (chain+reservoirs) Hamiltonian $H_{\text{tot}} = H + H_{\text{res}} + \epsilon H_I$ where $H_{\text{res}}$ is the reservoirs Hamiltonian, $H_I$ is the chain-reservoir coupling Hamiltonian and $\epsilon$ is a dimensionless parameter. If you abruptly turn on the coupling energy, the according coupling work is $W_I = \text{Tr}\,\rho_{\text{tot}}(H - H_0) = \epsilon\,\text{Tr}\,\rho_{\text{tot}}H_1$. This can be made arbitrarily small by decreasing $\epsilon$. Our assumption is that $\epsilon$ is so small that $W_I$ is negligible compared to the work done to attach/detach the end spins to the rest of the chain. Note that this may result in a slow relaxation, hence long thermalization time, meaning very low power. These are all aspects that need to be taken into account in practical realisations of the machine. Note also that in our cycle heat and work exchanges never occur simultaneously. While that condition is not necessary in order to have a well defined thermodynamic cycle (one could, for instance, devise cycles where thermalization and free evolution of the chain take place at the same time), that allows for a clear accounting of the energy balance. Thus any energy exchange occurring when the hot (H) or cold bath (C) is connected must be attributed entirely to heat dumped to the H bath, or the C respectively:

$$Q_H := \text{Tr}[H_A\,(\rho_A - \rho_A(\beta_1))], \qquad Q_C := \text{Tr}[H_B\,(\tilde{\rho}_B - \rho_B(\beta_2))], \tag{6}$$

where $\rho_A := \text{Tr}_{BC}[\rho_{ACB}]$ and $\tilde{\rho}_B := \text{Tr}_{BC}[\tilde{\rho}_{ACB}]$ are the reduced density matrices of $A$ and $B$ just before the thermalization events (i.e., at the beginning of the stokes 1 and 3 of the cycle).

Similarly, when the system is not in touch with any reservoirs, i.e., during the unitary evolution, no heat exchanges are possible. Hence, the work extracted in a cycle can be again evaluated by looking at the changes in internal energy during the quenches relative to connecting/disconnecting sites from the chain:

$$W_1 := \text{Tr}[H_{AC}\rho_{ACB}], \qquad W_2 := -\text{Tr}[H_{AC}\,(\rho_A(\beta_1) \otimes \rho_{CB})], \tag{7}$$

$$W_3 := \text{Tr}[H_{CB}\tilde{\rho}_{ACB}], \qquad W_4 := -\text{Tr}[H_{CB}\,(\tilde{\rho}_{AC} \otimes \rho_B(\beta_2))], \tag{8}$$

leading to a total work extracted from the system

$$W := W_1 + W_2 + W_3 + W_4. \tag{9}$$

The Von-Neumann entropy increase of the chain during the cycle can instead be computed as

$$\Delta S := S(\tilde{\rho}_{AC} \otimes \rho_B(\beta_2)) - S(\rho_{ACB}) = \Delta S_{T_1} + \Delta S_{T_2}, \tag{10}$$

with $\Delta S_{T_1}$ and $\Delta S_{T_2}$ the jumps associated to the thermalization events which obey the bounds

$$\Delta S_{T_1} := S(\rho_A(\beta_1) \otimes \rho_{CB}) - S(\rho_{ACB}) \geq S(\rho_A(\beta_1)) - S(\rho_A) \geq -\beta_1 Q_H, \tag{11}$$

$$\Delta S_{T_2} := S(\tilde{\rho}_{AC} \otimes \rho_B(\beta_2)) - S(\tilde{\rho}_{ACB}) \geq S(\rho_B(\beta_2)) - S(\tilde{\rho}_B) \geq -\beta_2 Q_C, \tag{12}$$

The first inequalities follows from the sub-additivity of the von Neumann entropy. The last inequalities follows from noticing that $S(\rho_A(\beta_1)) - S(\rho_A) + \beta_1 Q_H = S(\rho_A || \rho_A(\beta_1))$, and $S(\rho_B(\beta_2)) - S(\tilde{\rho}_B) + \beta_2 Q_C = S(\rho_B || \rho_B(\beta_2))$, where the non negative quantity $S(\rho || \sigma) = \text{Tr} \, \rho \ln \rho - \text{Tr} \, \rho \ln \sigma \geq 0$ denotes the Kullback-Liebler divergence (often referred to as relative entropy).

A case of particular interest is represented by those scenarios where after a cycle the chain returns to its original configuration, i.e.,

$$\mathcal{U}_2 \left( \tilde{\rho}_{AC} \otimes \rho_B(\beta_2) \right) = \rho_{ACB}, \tag{13}$$

a condition that is exhibited, e.g., when the model admits a limit cycle (see Sec. II B). When this happens then by construction the energy balance at the end of a cycle, reads, in accordance with the first principle of thermodynamics,

$$Q_H + Q_C + W = 0. \tag{14}$$

Similarly, under the condition (13) we get $\Delta S = 0$ which via (10)-(12), implies positivity of the "Clausius sum" – this is the discrete version of Clausius' celebrated inequality $\int \delta Q \leq 0$ [37](note the different sign convention for heat adopted here, though). We refrain from referring to the non-equilibrium Clausius sum $\Delta \mathcal{C}$ as an entropy change, because the change in thermodynamic entropy is in fact a lower bound for it.

$$\Delta \mathcal{C} := \beta_1 Q_H + \beta_2 Q_C \geq 0 , \tag{15}$$

in agreement with the second principle of thermodynamics. An aftermath of Eqs. (14,15) is that only four operating regimes are possible, depending on the relative signs of $Q_H$, $Q_C$ and $W$ [21–23]:

- For $Q_H < 0$, $Q_C > 0$, and $W > 0$, the machine acts as a conventional **thermal engine [E]** that at each cycle produces positive work by moving heat from the hot bath to the cold bath; the associated efficiency being bounded by the Carnot limit

$$\eta := \frac{W}{|Q_H|} \leq 1 - \frac{\beta_1}{\beta_2} , \tag{16}$$

 thanks to (14) and (15).

- For $Q_H > 0$, $Q_C < 0$, and $W < 0$, the machine acts as a conventional **refrigerator [R]** which extracts heat from the cold bath by absorbing external works and dissipating part of it into the hot bath; the coefficient of performances fulfilling the Carnot inequality

$$COP := \frac{|Q_C|}{|W|} \leq \frac{\beta_1}{\beta_2 - \beta_1} , \tag{17}$$

 thanks again to (14) and (15).

- For $Q_H < 0$, $Q_C > 0$, and $W < 0$ heat from the hot bath is moved into the cold bath while using external work: under this condition the machine behaves as a thermal **accelerator [A]**;

- For $Q_H > 0$, $Q_C > 0$, and $W < 0$ the machine operates instead as a **heater [H]** which converts external work into heat that is dumped in both baths.

### A. System dynamics

Given the state $\rho_{CB}^{(n)}$ of the subsystem $CB$ at the end of the stroke 1 of the $n$-th cycle, its state $\rho_{CB}^{(n+1)}$ after one complete 4 strokes cycle reads:

$$\rho_{CB}^{(m+1)} := \Phi_{CB}(\rho_{CB}^{(m)}) = \text{Tr}_A \left[ \mathcal{U}_2 \circ \mathcal{T}_2 \circ \mathcal{U}_1 \left( \rho_A(\beta_1) \otimes \rho_{CB}^{(m)} \right) \right] = \sum_{\alpha, \alpha'} R_{CB}^{(\alpha, \alpha')} \rho_{CB}^{(m)} R_{CB}^{(\alpha, \alpha')\dagger}. \tag{18}$$

Here the symbol "◦" denotes super-operator concatenation and

$$R_{CB}^{(\alpha,\alpha')} := \frac{e^{-\frac{\beta_1 \epsilon_i^A + \beta_2 \epsilon_{j'}^B}{2}}}{\sqrt{Z_A(\beta_1)Z_B(\beta_2)}} \, {}_A\langle j|U(\tau_2)|i\rangle_B \langle i'|U(\tau_1)|j'\rangle_A, \tag{19}$$

where $\alpha = (i,j)$ is a collective index, and $\epsilon_i^A$ and $|i\rangle_A$ (resp. $\epsilon_j^B$ and $|j\rangle_B$) are the energy eigenvalues and eigenvectors of the local Hamiltonian $H_A$ (resp. $H_B$). Note that we introduced the symbol $\Phi_{CB}$ to denote the map that evolves the system density matrix at the end of stroke 1 of one cycle. It is of crucial to observe that such map is a linear, completely positive, trace preserving (LCPTC) map, or, in short a so called quantum channel [35]. The operators $R_{CB}^{(\alpha,\alpha')}$ represent the Kraus operators associated to the LCPTC map $\Phi_{CB}$.

By iteration we can hence write

$$\rho_{CB}^{(m+1)} = \Phi_{CB}^m(\rho_{CB}^{(1)}) := \underbrace{\Phi_{CB} \circ \Phi_{CB} \circ \cdots \circ \Phi_{CB}}_{m \text{ times}}(\rho_{CB}^{(1)}) \,. \tag{20}$$

Similarly, indicating with $\tilde{\rho}_{AC}^{(m)}$ the state of the subsystem $AC$ at the end of the stroke 3 of the $m$-th cycle we can write

$$\begin{aligned} \tilde{\rho}_{AC}^{(m+1)} &= \tilde{\Phi}_{AC}(\tilde{\rho}_{AC}^{(m)}) := \text{Tr}_B\left[\mathcal{U}_1 \circ \mathcal{T}_1 \circ \mathcal{U}_2\left(\tilde{\rho}_{AC}^{(m)} \otimes \rho_B(\beta_2)\right)\right] = \sum_{\alpha,\alpha'} R_{AC}^{(\alpha,\alpha')} \tilde{\rho}_{AC}^{(m)} R_{AC}^{(\alpha,\alpha')\dagger} \\ &= \tilde{\Phi}_{AC}^m(\tilde{\rho}_{AC}^{(1)}) \,, \end{aligned} \tag{21}$$

where now $\tilde{\Phi}_{AC}$ is a quantum channel with Kraus operators

$$R_{AC}^{(\alpha,\alpha')} := \frac{e^{-\frac{\beta_1 \epsilon_i^A + \beta_2 \epsilon_{j'}^B}{2}}}{\sqrt{Z_A(\beta_1)Z_B(\beta_2)}} \, {}_B\langle j|U(\tau_1)|i\rangle_A \langle i'|U(\tau_2)|j'\rangle_B. \tag{22}$$

A case of particular interest is obtained when either $\tau_2 = 0$ or $\tau_1 = 0$, that is when the system becomes a two-strokes engine where a simultaneous thermalization of the first and last qubit is followed by a free evolution of the chain for a time $\tau$. In such scenario, after the thermalizations, the system is described by a state of the form $\rho_A(\beta_1) \otimes \rho_C \otimes \rho_B(\beta_2)$, and thus Eqs. (18) and (21) simplify to a map $\Phi_C$ that acts only on the subsystem system $C$ alone, i.e.

$$\begin{aligned} \rho_C^{(m+1)} &= \Phi_C(\rho_C^{(m)}) := \text{Tr}_{AB}\left[\mathcal{U}\left(\rho_A(\beta_1) \otimes \rho_C^{(m)} \otimes \rho_B(\beta_2)\right)\right] = \sum_{\alpha,\alpha'} R_C^{(\alpha,\alpha')} \rho_C^{(m)} R_C^{(\alpha,\alpha')\dagger} \\ &= \Phi_C^m(\rho_C^{(1)}) \,, \end{aligned} \tag{23}$$

with $\mathcal{U}$ the unitary super-operator associated with $U(\tau)$ and with

$$R_C^{(\alpha,\alpha')} := \frac{e^{-\frac{\beta_1 \epsilon_i^A + \beta_2 \epsilon_j^B}{2}}}{\sqrt{Z_A(\beta_1)Z_B(\beta_2)}} \, {}_{AB}\langle i',j'|U(\tau)|i,j\rangle_{AB}. \tag{24}$$

## B. Limit cycle

In the study of the asymptotic performance of the thermal machine, some important simplifications arise when the channels $\Phi_{CB}$, and $\tilde{\Phi}_{AC}$ are mixing [38, 39]. We recall that a quantum map $\Phi$ is said to be mixing when, irrespective of the initial condition $\rho$, the iterated application of $\Phi$ will drive the system toward a fixed point $\rho^{(\star)}$ represented by the unique configuration that is left invariant by the map, i.e.

$$\lim_{m\to\infty} \Phi^m(\rho) = \tilde{\rho}^{(\star)}, \qquad \Phi(\rho^{(\star)}) = \rho^{(\star)}. \tag{25}$$

Mixing channels form a dense set on the space of quantum transformations [38] while the set of non-mixing maps is a zero measure set. Physically this means that while mixing channels are the norm, non-mixing channels are the exception and any arbitrary small random perturbation is sufficient to make them mixing.

For the model we are studying here this translates into the fact that for almost any choice of $\tau_1$ and $\tau_2$, $\Phi_{BC}$ and $\tilde{\Phi}_{AC}$ will obey such property with fixed points $\rho_{CB}^{(\star)}$ and $\tilde{\rho}_{AC}^{(\star)}$ connected by the identities

$$\rho_{CB}^{(\star)} = \mathrm{Tr}_A \left[ \mathcal{T}_1 \circ \mathcal{U}_2 \left( \tilde{\rho}_{AC}^{(\star)} \otimes \rho_B(\beta_2) \right) \right] , \qquad \tilde{\rho}_{AC}^{(\star)} = \mathrm{Tr}_B \left[ \mathcal{T}_2 \circ \mathcal{U}_1 \left( \rho_A(\beta_1) \otimes \rho_{CB}^{(\star)} \right) \right] . \tag{26}$$

For the magnetization preserving spin-chain Hamiltonian model of Sec. III, a proof of this fact is explicitly shown in Appendix A. Physically, the mixing property of $\Phi_{CB}$ and $\tilde{\Phi}_{AC}$ implies that independently from the system initialization, as $m$ increases, our thermodynamical cycle will be driven toward the following limit cycle,

$$\rho_A(\beta_1) \otimes \rho_{CB}^{(\star)} \qquad \xrightarrow{\text{(stroke 2)}} \qquad \tilde{\rho}_{ACB}^{(\star)} := U(\tau_1) \left( \rho_A(\beta_1) \otimes \rho_{CB}^{(\star)} \right) U^\dagger(\tau_1)$$

$$\uparrow_{\text{(stroke 1)}} \qquad\qquad\qquad\qquad\qquad\qquad \downarrow_{\text{(stroke 3)}} \tag{27}$$

$$\rho_{ACB}^{(\star)} := U(\tau_2) \left( \tilde{\rho}_{AC}^{(\star)} \otimes \rho_B(\beta_2) \right) U^\dagger(\tau_2) \qquad \xleftarrow{\text{(stroke 4)}} \qquad \tilde{\rho}_{AC}^{(\star)} \otimes \rho_B(\beta_2) ,$$

which fulfils the close loop condition (13) enabling us to invoke the identities (14)–(15) [notice that in the above expression $\rho_{CB}^{(\star)} := \mathrm{Tr}_A[\rho_{ACB}^{(\star)}]$, $\tilde{\rho}_{AC}^{(\star)} := \mathrm{Tr}_B[\tilde{\rho}_{ACB}^{(\star)}]$ corresponds to the reduced density operators of $\rho_{ACB}^{(\star)}$ and $\tilde{\rho}_{ACB}^{(\star)}$ respectively]. Under this condition the average heat exchanges and work produced per cycle can be computed in terms of the corresponding quantities associated with the limit cycle, i.e.,

$$\overline{Q}_H := \lim_{M \to \infty} \frac{\sum_{m=1}^N Q_H^{(m)}}{M} = \lim_{M \to \infty} \frac{\sum_{m=1}^M \mathrm{Tr}\left[ H_A \left( \rho_A^{(m)} - \rho_A(\beta_1) \right) \right]}{M}$$
$$= \mathrm{Tr}\left[ H_A \left( \rho_A^{(\star)} - \rho_A(\beta_1) \right) \right] =: Q_H^{(\star)} , \tag{28}$$

$$\overline{Q}_C := \lim_{M \to \infty} \frac{\sum_{m=1}^N Q_C^{(m)}}{M} = \lim_{M \to \infty} \frac{\sum_{m=1}^M \mathrm{Tr}\left[ H_B \left( \tilde{\rho}_B^{(m)} - \rho_B(\beta_2) \right) \right]}{M}$$
$$= \mathrm{Tr}\left[ H_B \left( \tilde{\rho}_B^{(\star)} - \rho_B(\beta_2) \right) \right] =: Q_C^{(\star)} , \tag{29}$$

$$\overline{W} := \lim_{M \to \infty} \frac{\sum_{m=1}^M W^{(m)}}{M} = -(Q_H^{(\star)} + Q_C^{(\star)}) =: W^{(\star)} , \tag{30}$$

$$\overline{\Delta \mathcal{C}} := \lim_{M \to \infty} \frac{\sum_{m=1}^M (\beta_1 Q_H^{(m)} + \beta_2 Q_C^{(m)})}{M} = \beta_1 Q_H^{(\star)} + \beta_2 Q_C^{(\star)}) =: \Delta \mathcal{C}^{(\star)} , \tag{31}$$

with $\rho_A^{(\star)} := \mathrm{Tr}_{CB}[\rho_{ACB}^{(\star)}]$ and $\tilde{\rho}_B^{(\star)} := \mathrm{Tr}_{AC}[\tilde{\rho}_{ACB}^{(\star)}]$ being, respectively, the reduced density matrices of $A$ and $B$ at beginning of stroke 1 and 3 of the limit cycle. Notice finally that in the two-stroke regime (22) the loop (27) reduces to

$$\rho_A(\beta_1) \otimes \rho_C^{(\star)} \otimes \rho_B(\beta_2) \ \rightleftarrows \ U(\tau) \left( \rho_A(\beta_1) \otimes \rho_C^{(\star)} \otimes \rho_B(\beta_2) \right) U^\dagger(\tau) , \tag{32}$$

with $\rho_C^{(\star)}$ the fixed point state of $\Phi_C$, while Eqs. (28)–(30) hold true by identifying $\rho_A^{(\star)}$ and $\tilde{\rho}_B^{(\star)}$ with reduced density matrices of $U(\tau) \left( \rho_A(\beta_1) \otimes \rho_C^{(\star)} \otimes \rho_B(\beta_2) \right) U^\dagger(\tau)$.

## III.  MAGNETIZATION PRESERVING SPIN-CHAIN MODELS

In this section, we analyse the case where subsystems $A$ and $B$ are the first and last element of a linear chain of $N(\geq 2)$ spin-1/2 particles coupled with an Hamiltonian that preserves the total magnetization along the longitudinal $z$-axis $S^Z := \sum_{i=1}^N S_i^Z$, i.e.,

$$H = \sum_{i=1}^N E_i S_i^Z + \sum_{i=1}^{N-1} 4J_i \left( S_i^X S_{i+1}^X + S_i^Y S_{i+1}^Y \right) + \sum_{i=1}^{N-1} 4K_i \left( S_i^X S_{i+1}^Y - S_i^Y S_{i+1}^X \right) + \sum_{i=1}^{N-1} 4F_i S_i^Z S_{i+1}^Z , \tag{33}$$

where for $i = 1, \cdots, N$, $S_i^{X,Y,Z}$ represent the $X$, $Y$, $Z$ spin operators of the $i$-th particle of the model, and where the real parameters $E_i$ define the local energy terms, $J_i$, $K_i$ spin exchange terms, and $F_i$ the standard Ising coupling

terms. As anticipated in the previous section, for almost any choice of the system parameters the quantum maps $\Phi_{CB}$ and $\tilde{\Phi}_{AC}$ of the model exhibit mixing properties. This fact is explicitly proved in Appendix A where we also show that the associated fixed point states (26) commute with the associated magnetization operators $S^Z_{CB} := \sum_{i=2}^N S^Z_i$ and $S^Z_{AC} := \sum_{i=1}^{N-1} S^Z_i$, i.e.,

$$\begin{cases} \Phi_{CB} \text{ mixing} & \& \quad \Phi_{CB}(\rho^{(\star)}_{CB}) = \rho^{(\star)}_{CB} & \implies & [\rho^{(\star)}_{CB}, S^Z_{CB}] = 0 \,, \\ \tilde{\Phi}_{AC} \text{ mixing} & \& \quad \tilde{\Phi}_{AC}(\tilde{\rho}^{(\star)}_{AC}) = \tilde{\rho}^{(\star)}_{AC} & \implies & [\tilde{\rho}^{(\star)}_{AC}, S^Z_{AC}] = 0 \,. \end{cases} \tag{34}$$

Our analysis will target the case of mixing maps only, by analyzing the performance of the according limit . Under this conditions, in Sec. III A will shall prove that the operation regimes of the machine are fully determined by local properties of the sites $A$ and $B$. In Sec. III B we will then proceed with the explicit evaluation of the heat exchanges of the limit cycle.

## A. Operation regimes

In the study of the limit cycles of magnetization preserving spin-chains few important simplifications arise.
First of all it can be shown that the following implication holds:

$$\beta_1 E_1 = \beta_2 E_N \quad \implies \quad Q^{(\star)}_H = Q^{(\star)}_C = 0 \,. \tag{35}$$

Indeed, indicating with $S^Z_C := \sum_{i=2}^{N-1} S^Z_i$ the longitudinal magnetization of the $C$ part of the spin-chain, due to the fact that $H$ and $S^Z$ commute, one can verify that for $\kappa = \beta_1 E_1 = \beta_2 E_N$ the state

$$\rho_A(\beta_1) \otimes \frac{e^{-\kappa S^Z_C}}{\text{Tr}[e^{-\kappa S^Z_C}]} \otimes \rho_B(\beta_2) = \frac{e^{-\kappa S^Z}}{\text{Tr}[e^{-\kappa S^Z}]} \,, \tag{36}$$

is invariant under the transformations $\mathcal{U}_2 \circ \mathcal{T}_1 \circ \mathcal{U}_1$. Accordingly, Eq. (18) then forces us to identify the associated reduced density matrix w.r.t. to $BC$ as the (unique) fixed point of the mixing channel $\Phi_{CB}$, leading to the condition

$$\rho^{(\star)}_{CB} = \text{Tr}_A \left[ \frac{e^{-\kappa S^Z}}{\text{Tr}[e^{-\kappa S^Z}]} \right] \quad \implies \quad \rho^{(\star)}_B = \text{Tr}_{AC} \left[ \frac{e^{-\kappa S^Z}}{\text{Tr}[e^{-\kappa S^Z}]} \right] = \rho_B(\beta_2) \,, \tag{37}$$

which via Eq. (29) finally leads to $Q^{(\star)}_C = 0$ – the proof that $Q^{(\star)}_H = 0$ follows by the same reasoning noticing that (36) is also invariant under $\mathcal{U}_1 \circ \mathcal{T}_2 \circ \mathcal{U}_2$ and invoking (21) and (28).

The second important simplification that applies to the models (33) is that, as schematically shown in Fig. 2, the operation regimes [H], [R], [E], and [A] can be directly linked to the ratio between the local energy parameters $E_1$ and $E_N$ of $A$ and $B$, via the following simple rules

$$\frac{E_N}{E_1} < 0 \quad \implies \quad [\text{H}] \,, \quad 0 < \frac{E_N}{E_1} < \frac{\beta_1}{\beta_2} \quad \implies \quad [\text{R}] \,,$$

$$\frac{\beta_1}{\beta_2} < \frac{E_N}{E_1} < 1 \quad \implies \quad [\text{E}] \,, \qquad \frac{E_N}{E_1} > 1 \quad \implies \quad [\text{A}] \,. \tag{38}$$

Equation (38) stems directly from the fact that for Hamiltonians of the form (33) the heat exchanges of the limit cycle can be related by the following identity:

$$\frac{Q^{(\star)}_H}{E_1} + \frac{Q^{(\star)}_C}{E_N} = 0 \,. \tag{39}$$

Such a symmetry is a direct consequence of the fact that the total magnetisation along Z is a conserved quantity (an example of spin-chain model with not commuting $H$ and $S$ which do not satisfy (39) will be presented in Sec. IV). To see this explicitly observe that since $U(\tau_1)$ and $U(\tau_2)$ preserves the total longitudinal magnetization, from Eq. (27) it follows that

$$\text{Tr}\left[S^Z \tilde{\rho}^{(\star)}_{ACB}\right] = \text{Tr}\left[S^Z U(\tau_1) \left(\rho_A(\beta_1) \otimes \rho^{(\star)}_{CB}\right) U^\dagger(\tau_1)\right] = \text{Tr}\left[S^Z \left(\rho_A(\beta_1) \otimes \rho^{(\star)}_{CB}\right)\right] \,, \tag{40}$$

$$\text{Tr}\left[S^Z \rho^{(\star)}_{ACB}\right] = \text{Tr}\left[S^Z U(\tau_2) \left(\tilde{\rho}^{(\star)}_{AC} \otimes \rho_B(\beta_2)\right) U^\dagger(\tau_2)\right] = \text{Tr}\left[S^Z \left(\tilde{\rho}^{(\star)}_{AC} \otimes \rho_B(\beta_2)\right)\right] \,. \tag{41}$$

Indicating with $S_D^Z$ the logintudinal magnetization of the segment $D$ of the chain, and writing $S^Z = S_A^Z + S_C^Z + S_B^Z$ we can now transform the above expressions in the following identities

$$\mathrm{Tr}_{AC}[(S_A^Z + S_C^Z)\tilde{\rho}_{AC}^{(\star)}] + \mathrm{Tr}_B[S_B^Z\tilde{\rho}_B^{(\star)}] = \mathrm{Tr}_A[S_A^Z\rho_A(\beta_1)] + \mathrm{Tr}_{CB}[(S_C^Z + S_B^Z)\rho_{CB}^{(\star)}] , \tag{42}$$

$$\mathrm{Tr}_A[S_B^Z\rho_A^{(\star)}] + \mathrm{Tr}_{CB}[(S_C^Z + S_B^Z)\rho_{CB}^{(\star)}] = \mathrm{Tr}_{AC}[(S_A^Z + S_C^Z)\tilde{\rho}_{AC}^{(\star)}] + \mathrm{Tr}_B[S_B^Z\rho_B(\beta_2)] , \tag{43}$$

which substracted term by term, finally lead to

$$\mathrm{Tr}_A\left[S_A^Z(\rho_A^{(\star)} - \rho_A(\beta_1))\right] + \mathrm{Tr}_B\left[S_B^Z(\rho_B^{(\star)} - \rho_B(\beta_2))\right] = 0 , \tag{44}$$

that corresponds to (39) thanks to the fact that $H_A = E_1 S_A^Z$ and $H_B = E_N S_B^Z$. The derivation of (38) then follows by using (39) to rewrite Eqs. (14), (15) as

$$W^{(\star)} = -(Q_H^{(\star)} + Q_C^{(\star)}) = -Q_H^{(\star)}\left(1 - \frac{E_N}{E_1}\right) , \tag{45}$$

$$\Delta\mathcal{C}^{(\star)} = \beta_1 Q_H^{(\star)} + \beta_2 Q_C^{(\star)} = \beta_2 Q_H^{(\star)}\left(\frac{\beta_1}{\beta_2} - \frac{E_N}{E_1}\right) \geq 0 . \tag{46}$$

Notice in fact that from Eq. (45) we get that $W^{(\star)}$ and $Q_H^{(\star)}$ can have the same sign if and only if $\frac{E_N}{E_1} \leq 1$, while Eq. (46) imposes the conditions

$$\begin{cases} \frac{E_N}{E_1} > \frac{\beta_1}{\beta_2} & \Leftrightarrow \quad Q_H^{(\star)} \leq 0 , \ Q_C^{(\star)} \geq 0 , \\ \\ \frac{E_N}{E_1} < \frac{\beta_1}{\beta_2} & \Leftrightarrow \quad Q_H^{(\star)} \geq 0 , \ Q_C^{(\star)} \leq 0 . \end{cases} \tag{47}$$

(for $\frac{E_N}{E_1} = \frac{\beta_1}{\beta_2}$ no definite sign can be assigned to $Q_H^{(\star)}$, $Q_C^{(\star)}$). A close inspection of the above relations reveals that indeed as predicted in Eq. (40), for $\frac{E_N}{E_1} < 0$ the system behaves as a heater [H], while for $\frac{E_N}{E_1} > 1$ it behaves as a thermal accelerator [A]. On the contrary for $0 < \frac{E_N}{E_1} < \frac{\beta_1}{\beta_2}$ the chain operates as a refrigerator [R] and for $\frac{\beta_1}{\beta_2} < \frac{E_N}{E_1} < 1$ as a thermal engine regime [E]. In these latter two cases, Eqs. (45) and (39) also fully determine the associated efficiencies with formulas

$$COP = \frac{|Q_C^{(\star)}|}{|W^{(\star)}|} = \frac{E_N}{E_1 - E_N} , \qquad \eta = \frac{|W^{(\star)}|}{|Q_H^{(\star)}|} = 1 - \frac{E_N}{E_1} , \tag{48}$$

that closely resemble those one would get for a two-level-system quantum Otto engine or a for a two-stroke two-qubit engine [21, 23, 40].

## B. Evaluating the heat exchanges

According to Eqs. (28)–(31) to compute the energy exchanges of the limit cycle one needs to determine the fixed point state $\rho_{CB}^{(\star)}$ of the map $\Phi_{BC}$. Such task can be approached analytically only for very small chains ($N \leq 3$) or under special assumptions on the system settings. Nonetheless from our analysis it emerges a universal behaviour that we formalize in the following ansatz:

$$\begin{cases} Q_C^{(\star)} = g(\beta_1 E_1, \beta_2 E_N)f_4(\tau_1, \tau_2)E_N , \\ Q_H^{(\star)} = -g(\beta_1 E_1, \beta_2 E_N)f_4(\tau_1, \tau_2)E_1 , \\ W^{(\star)} = g(\beta_1 E_1, \beta_2 E_N)f_4(\tau_1, \tau_2)(E_1 - E_N) , \\ \Delta\mathcal{C}^{(\star)} = g(\beta_1 E_1, \beta_2 E_N)f_4(\tau_1, \tau_2)(\beta_2 E_N - \beta_1 E_1) , \end{cases} \tag{49}$$

where

$$g(\beta_1 E_1, \beta_2 E_N) := \frac{e^{\beta_2 E_N} - e^{\beta_1 E_1}}{(e^{\beta_2 E_N} + 1)(e^{\beta_1 E_1} + 1)} , \tag{50}$$

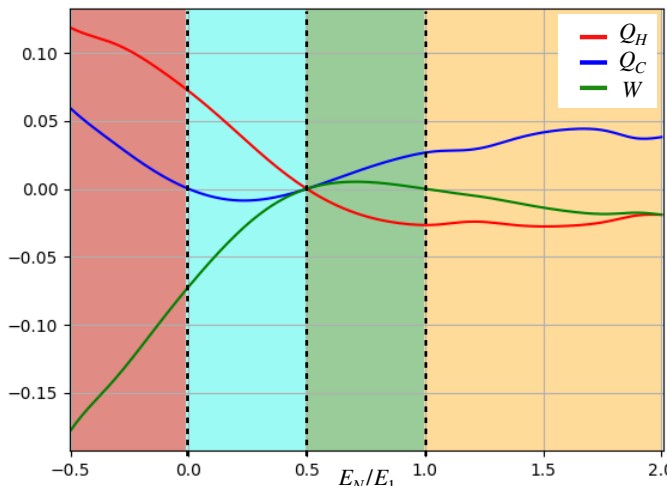

FIG. 2: Operation regimes of the chain as a function of the parameter $E_N/E_1$ as established in Sec. III: Red area heater [H], light blue refrigerator [R], green thermal engine [E], yellow thermal accelerator [A]. The red, blue and green curves represents the values of $Q_H^{(\star)}$, $Q_C^{(\star)}$ and $W^{(\star)}$ as computed in Sec. III B 3. In the plot we assume $N = 8$ spins and fixed the temperature ratio $\beta_1/\beta_2 = 0.5$. $Q_H, Q_C, W$ are expressed in units of $E_1$.

and where for all the quantities, the dependence upon the free evolution time intervals $\tau_1$ and $\tau_2$ is carried out by one and the same (model dependent) function $f_4$ that is independent of the bath temperatures and fulfils the property

$$0 \leq f_4(\tau_1, \tau_2) \leq 1 \ . \tag{51}$$

Notice that the last two identities in Eq. (49) are just a direct consequence of the first two and of Eqs. (45) and (46), and that the entire set of equations can also be applied for the special case of the two-strokes cycle of Eq. (23) by replacing $f_4(\tau_1, \tau_2)$ with $f_2(\tau) := f_4(\tau, 0) = f_4(0, \tau)$ (the symmetry between $f_4(\tau, 0)$ and $f_4(0, \tau)$ being a direct consequence of the fact that the 2 stokes machine can be equivalently obtained either setting $\tau_1$ or $\tau_2$ to zero). We remark finally that Eqs. (49)–(51) assign values to $Q_C^{(\star)}$, $Q_H^{(\star)}$ and $W^{(\star)}$ which are in perfect agreement with the regimes (38) established in Sec. III A.

While a general proof of Eq. (49) for all possible choices of the Hamiltonian (33) is still missing, in the subsequent sections we report analytical and numerical evidences that suggest that this is indeed the case. We stress that being able to establish the validity of the ansatz above would allow one to enormously simplify the study of the model reducing it to just the determination of the function $f_4$, a task which thanks to the fact that such term does not depend upon $T_1$ and $T_2$, can be easily accomplished through the low-temperature limit analysis reported in Sec. III B 2.

### 1. Small chain limit

The model allows for full analytical solution for a spin-chain of $N = 2$ elements. In this case the $C$ section of the chain is absent and $A$ and $B$ are directly connected by an Hamiltonian coupling of the form

$$H = E_1 S_1^Z + E_2 S_2^Z + 4J \left( S_1^X S_2^X + S_1^Y S_2^Y \right) + 4K \left( S_1^X S_2^Y - S_1^Y S_2^X \right) + F S_1^Z S_2^Z \ , \tag{52}$$

which can be formally obtained from (33) by identifying $C$ with (say) $B$. By direct computation it turns out that the unique fixed point of the map $\Phi_B$ is provided by the density operator $\rho_B^{(\star)}$ that, in agreement with Observation 2 of App. A, is diagonal in the eigenbasis $\{|0\rangle_B, |1\rangle_B\}$ of the magnetization operator $S_B^Z$ with diagonal entries

$$_B\langle 1|\rho_B^{(\star)}|1\rangle_B = \frac{\frac{e^{-\frac{\beta_1 E_1}{2}}}{Z_1(\beta_1)}|S_H(\tau_2)|^2|C_H(\tau_1)|^2 + \frac{e^{-\frac{\beta_2 E_2}{2}}}{Z_2(\beta_2)}|C_H(\tau_2)|^2}{1 - |S_H(\tau_1)|^2|S_H(\tau_2)|^2} \ , \tag{53}$$

$$_B\langle 0|\rho_B^{(\star)}|0\rangle_B = \frac{\frac{e^{\frac{\beta_1 E_1}{2}}}{Z_1(\beta_1)}|S_H(\tau_2)|^2|C_H(\tau_1)|^2 + \frac{e^{\frac{\beta_2 E_2}{2}}}{Z_2(\beta_2)}|C_H(\tau_2)|^2}{1 - |S_H(\tau_1)|^2|S_H(\tau_2)|^2} \ , \tag{54}$$

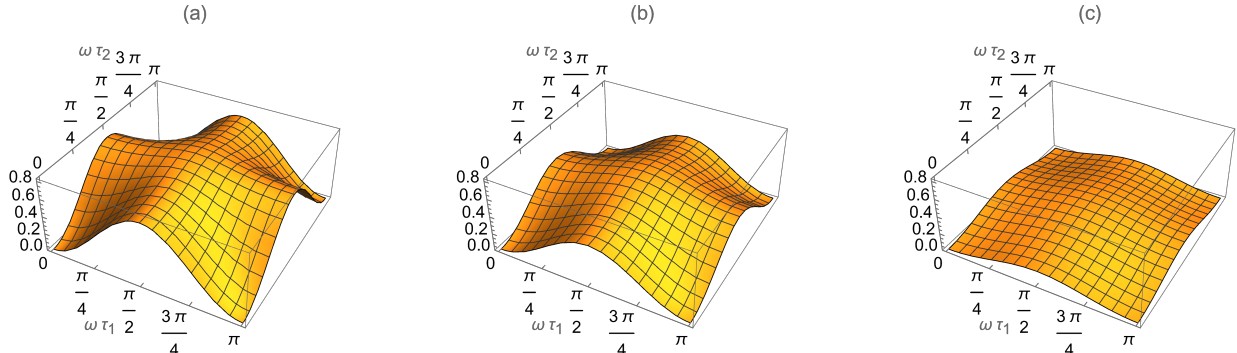

FIG. 3: Plot of the function $f_4(\tau_1, \tau_2)$ of Eq. (58) as a function of $\tau_1$ and $\tau_2$ for three values of $W^2/(4\omega^2)$ – explicitly 0.7 (a), 0.45 (b), and 0.15 (c).

where

$$C_H(\tau) := \cos(\omega\tau) + i\sin(\omega\tau)\frac{E_2 - E_1}{2\omega} , \tag{55}$$

$$S_H(\tau) := -\sin(\omega\tau)\frac{J - iK}{2\omega} , \tag{56}$$

$$\omega := \frac{\sqrt{(E_2 - E_1)^2 + |J|^2 + |K|^2}}{2} . \tag{57}$$

Replacing this into Eqs. (28)–(31) allows to express the thermodynamic quantities of the model as in Eq. (49)

$$f_4(\tau_1, \tau_2) = \frac{W^2}{4\omega^2}\frac{\sin^2(\omega\tau_1) + \sin^2(\omega\tau_2) - 2\frac{W^2}{4\omega^2}\sin^2(\omega\tau_1)\sin^2(\omega\tau_2)}{1 - \frac{W^4}{16\omega^4}\sin^2(\omega\tau_1)\sin^2(\omega\tau_2)} , \tag{58}$$

where $W^2 := J^2 + K^2$ (see Fig. 3).

Other cases which we solved analytically are the two-strokes cycles (23) of spin-chains of $N = 3$ elements with Hamiltonian

$$H = \sum_{i=1}^{N} E_i S_i^Z + \sum_{i=1}^{N-1} 4J_i[S_i^X S_{i+1}^X + S_i^Y S_{i+1}^Y] . \tag{59}$$

In this scenario we find it convenient to express the unitary operator $U(\tau)$ in the longitudinal magnetization basis where it assumes the following block-diagonal form

$$U(\tau) = \begin{pmatrix} U^{(3)}(\tau) & 0 & 0 & 0 \\ 0 & \boxed{U^{(2)}(\tau)} & 0 & 0 \\ 0 & 0 & \boxed{U^{(1)}(\tau)} & 0 \\ 0 & 0 & 0 & U^{(0)}(\tau) \end{pmatrix} , \tag{60}$$

with $U^{(j)}(\tau)$ representing the matrix associated with the subspace of $ACB$ that has $j$ spins up along the $z$-th direction. For the selected Hamiltonian one can show that the following constraint holds

$$\begin{cases} a(\tau) = |U_{12}^{(1)}(\tau)|^2 = |U_{12}^{(2)}(\tau)|^2 = |U_{21}^{(1)}(\tau)|^2 = |U_{21}^{(2)}(\tau)|^2 \\ b(\tau) = |U_{23}^{(1)}(\tau)|^2 = |U_{23}^{(2)}(\tau)|^2 = |U_{32}^{(1)}(\tau)|^2 = |U_{32}^{(2)}(\tau)|^2 \\ c(\tau) = |U_{31}^{(1)}(\tau)|^2 = |U_{31}^{(2)}(\tau)|^2 = |U_{13}^{(1)}(\tau)|^2 = |U_{13}^{(2)}(\tau)|^2 . \end{cases} \tag{61}$$

From this one can explicitly compute the fixed point $\rho_C^{(\star)}$ of the map $\Phi_C$ of Eq. (23) obtaining a density matrix which

in the local energy basis of $C$ has elements

$$
\begin{aligned}
{}_C\langle 1|\rho_C^{(\star)}|0\rangle_C &= 0 \,, \\
{}_C\langle 1|\rho_C^{(\star)}|1\rangle_C &= \frac{e_1^- e_2^- \left(|U_{12}^{(2)}|^2+|U_{32}^{(2)}|^2\right)+e_1^- e_2^+ |U_{21}^{(1)}|^2+e_1^+ e_2^- |U_{23}^{(1)}|^2}{e_1^- e_2^- \left(|U_{12}^{(2)}|^2+|U_{32}^{(2)}|^2\right)+e_1^- e_2^+ \left(|U_{21}^{(1)}|^2+|U_{23}^{(2)}|^2\right)+e_1^+ e_2^- \left(|U_{23}^{(1)}|^2+|U_{21}^{(2)}|^2\right)+e_1^+ e_2^+ \left(|U_{12}^{(1)}|^2+|U_{32}^{(1)}|^2\right)} \,, \\
{}_C\langle 0|\rho_C^{(\star)}|0\rangle_C &= \frac{e_1^- e_2^+ |U_{23}^{(2)}|^2+e_1^+ e_2^- |U_{21}^{(2)}|^2+e_1^+ e_2^+ \left(|U_{12}^{(1)}|^2+|U_{32}^{(1)}|^2\right)}{e_1^- e_2^- \left(|U_{12}^{(2)}|^2+|U_{32}^{(2)}|^2\right)+e_1^- e_2^+ \left(|U_{21}^{(1)}|^2+|U_{23}^{(2)}|^2\right)+e_1^+ e_2^- \left(|U_{23}^{(1)}|^2+|U_{21}^{(2)}|^2\right)+e_1^+ e_2^+ \left(|U_{12}^{(1)}|^2+|U_{32}^{(1)}|^2\right)} \,,
\end{aligned}
$$

(62)

where we set $e_1^\pm := e^{\pm \frac{\beta_1 E_1}{2}}$ and $e_2^\pm := e^{\pm \frac{\beta_2 E_3}{2}}$ in order to write the equations more compactly. With the help of the above expressions Eqs. (28)–(31) finally lead to the identities (49) with $f_2$ given by

$$
f_2(\tau) = \frac{a(\tau)b(\tau) + b(\tau)c(\tau) + c(\tau)a(\tau)}{a(\tau) + b(\tau)} \,.
$$

(63)

### 2. Low-temperature limit

The complete analytical evaluation of the energy exchanges at the limit cycle for a generic Hamiltonian is a hard task as the complexity of the problem grows exponentially with the size of the chain. One way to circumvent this issue is to focus on the limit in which the temperatures of the baths are much smaller than the energy gaps of the qubits with which they interact. We consequently define the small dimensionless constant $x_1 := e^{-\beta_1 E_1}$ and $x_2 := e^{-\beta_2 E_N}$ to determine the thermodynamic properties up to first order in such parameters. For the sake of simplicity, we shall also restrict the analysis to the two-strokes version of the model whose asymptotic performances are defined in Eq. (23): the result we obtain however can be generalized to the more general four-strokes scenario.

Indicating with $\Phi_C^{[0]}$ the LCPTP map $\Phi_C^{(0,0)}$ which describes the evolution of $C$ when both the baths of the model are initialized at zero temperature, following the derivation of Observation 3 of App. A, in the low-temperature limit, the first-order expansion in $x_j$ of the channel $\Phi_C$ can be written as

$$
\Phi_C = \Phi_C^{[0]} + \sum_{j=1}^2 x_j \Delta\Phi_C^{[j]} \,,
$$

(64)

$$
\Delta\Phi_C^{[1]} := \Phi_C^{(1,0)} - \Phi_C^{(0,0)} \,, \qquad \Delta\Phi_C^{[2]} := \Phi_C^{(0,1)} - \Phi_C^{(0,0)} \,,
$$

(65)

where as in Eq. (A6) (resp. (A7)) the map $\Phi_C^{(1,0)}$ ($\Phi_C^{(0,1)}$) represents scenarios where the bath of $A$ ($B$) inject exactly a single spin-up on the system while the bath $B$ ($A$) is still at zero-temperature. It is worth remarking that while being an approximation of the real limit cycle transformation of the model (it misses all the higher order contributions in $x_j$), the map (64) is a proper LCPTP channel as it is expressed as a convex combination of maps that have the same property. Accordingly to study its mixing properties we can apply to it the mathematical tools developed in [38, 39]. Furthermore, the same derivation given in Observation 1 of App. A can be used to show that, similarly to what happens in Eq. (34), if the channel (64) is mixing, then its fixed point state $\rho_C^{(\star)}$ must commute with the longitudinal magnetization operator $S_C^Z$.

To compute $\rho_C^{(\star)}$ we express it as a linear expansion in the $x_j$ parameters

$$
\rho_C^{(\star)} = \rho_C^{(\star)}[0] + \sum_{j=1}^2 x_j \Delta\rho_C^{(\star)}[j] \,,
$$

(66)

where the zero-th order term $\rho_C^{(\star)}[0]$ is the fixed point of $\Phi_C^{[0]}$ (i.e., the pure state $|0\cdots0\rangle_C$ where all the spins of the $C$ section are pointing down – see Observation 4 of App. A), while $\Delta\rho_C^{(\star)}[j]$ are operators that need to be determined. Replacing (64), (66) into the fixed point equation $\Phi_C(\rho_C^{(\star)}) = \rho_C^{(\star)}$, the first order terms in $x_j$ yield

$$
\Delta\rho_C^{(\star)}[j] = \Phi_C^{[0]}(\Delta\rho_C^{(\star)}[j]) + \Delta\Phi_C^{[j]}(\rho_C^{(\star)}[0]) \,,
$$

(67)

which upon $n$ iterations can be casted in the equivalent form

$$
\Delta\rho_C^{(\star)}[j] = (\Phi_C^{[0]})^n(\Delta\rho_C^{(\star)}[j]) + \sum_{k=0}^{n-1}(\Phi_C^{[0]})^k(\Delta\Phi_C^{[j]}(\rho_C^{(\star)}[0])) \,.
$$

(68)

Since the channel $\Phi_C^{[0]}$ is mixing with fixed point $\rho_C^{(\star)}[0]$, in the $n \to \infty$ limit the first contribution can be neglected due to the identity [38, 39]

$$\lim_{n \to \infty} (\Phi_C^{[0]})^n (\Delta \rho_C^{(\star)}[j]) = \mathrm{Tr}\left[\Delta \rho_C^{(\star)}[j]\right] \rho_C^{(\star)}[0] = 0 \,, \tag{69}$$

where we used the fact that $\Delta \rho_C^{(\star)}[j]$ must have zero trace in order to ensure the proper normalization of $\rho_C^{(\star)}$ – see Eq. (66). Accordingly we can replace (69) with

$$\Delta \rho_C^{(\star)}[j] = \sum_{k=0}^{\infty} (\Phi_C^{[0]})^k (\Delta \Phi_C^{[j]}(\rho_C^{(\star)}[0])) \,, \tag{70}$$

that expresses $\Delta \rho_C^{(\star)}[j]$ in terms of purely dynamical parameters of the model (i.e., the free evolution time $\tau$ and the Hamiltonian couplings). Observe that since $\rho_C^{(\star)}[0]$ is block diagonal (**bd**) with respect to the magnetization eigenbases decomposition of $C$, and since, as discussed in Observation 2 of App. A, $\Phi_C^{(0,0)}$, $\Phi_C^{(1,0)}$ and $\Phi_C^{(0,1)}$ are **bd** preserving transformations, it turns out that $\Delta \rho_C^{(\star)}[j]$ is also a **bd** operator, in agreement with the fact that (66) must commute with $S_C^Z$. Notice also that since $\Phi_C^{(1,0)}$ and $\Phi_C^{(0,1)}$ can at most increase by 1 the total number of spin-up in $C$, while $\Phi_C^{(0,0)}$ always tend to reduce it, Eq. (70) implies that $\Delta \rho_C^{(\star)}[j]$ can have components only on the magnetization eigenspaces with at most one single spin up in $C$. Specifically, reminding that $\rho_C^{(\star)}[0]$ is the unique non trivial operator in the subspace of $C$ which has no spin-up terms, we can write them as

$$\Delta \rho_C^{(\star)}[j] = \gamma_j (\varrho_C^{(\star)}[j] - \rho_C^{(\star)}[0]) \,, \tag{71}$$

with $\varrho_C^{(\star)}[j]$ being a density matrix of $C$ which contains exactly one excitation, and $\gamma_j \in [0,1]$ (indeed the possibility that $\varrho_C^{(\star)}[j]$ have negative eigenvalues as well as, the possibility to have $\gamma_j > 1$, are both ruled out by the positivity of $\rho_C^{(\star)}$).

Following (28)-(29) we can now compute the thermodynamic quantities of the limit cycle by determining the operators

$$\rho_A^{(\star)} - \rho_A(\beta_1) := \mathrm{Tr}_{CB}\left[\mathcal{U}(\tau)\left(\rho_A(\beta_1) \otimes \rho_C^{(\star)} \otimes \rho_B(\beta_2)\right)\right] - \rho_A(\beta_1) \,,$$
$$\rho_B^{(\star)} - \rho_B(\beta_2) := \mathrm{Tr}_{AC}\left[\mathcal{U}(\tau)\left(\rho_A(\beta_1) \otimes \rho_C^{(\star)} \otimes \rho_B(\beta_2)\right)\right] - \rho_B(\beta_2) \,, \tag{72}$$

which, using Eq. (66) and the fact that at first order one has $\rho_A(\beta_1) = |0\rangle_A\langle 0| + x_1(|1\rangle_A\langle 1| - |0\rangle_A\langle 0|)$, $\rho_B(\beta_2) = |0\rangle_B\langle 0| + x_2(|1\rangle_B\langle 1| - |0\rangle_B\langle 0|)$, can be written as

$$\rho_A^{(\star)} - \rho_A(\beta_1) = \sum_{j=1}^{2} x_j \Theta_A^{(j)}(\tau) \,, \qquad \rho_B^{(\star)} - \rho_B(\beta_2) = \sum_{j=1}^{2} x_j \Theta_B^{(j)}(\tau) \,, \tag{73}$$

with

$$\begin{cases} \Theta_A^{(1)}(\tau) := \mathrm{Tr}_{CB}\left[\mathcal{U}(\tau)\left(|00\rangle_{AB}\langle 00| \otimes \Delta\rho_C^{(\star)}[1]\right) + |10\cdots 0\rangle_{ACB}\langle 10\cdots 0|\right] - |1\rangle_A\langle 1| \,, \\ \Theta_A^{(2)}(\tau) := \mathrm{Tr}_{CB}\left[\mathcal{U}(\tau)\left(|00\rangle_{AB}\langle 00| \otimes \Delta\rho_C^{(\star)}[2]\right) + |0\cdots 01\rangle_{ACB}\langle 0\cdots 01|\right] - |0\rangle_A\langle 0| \,, \\ \Theta_B^{(1)}(\tau) := \mathrm{Tr}_{AC}\left[\mathcal{U}(\tau)\left(|00\rangle_{AB}\langle 00| \otimes \Delta\rho_C^{(\star)}[1]\right) + |10\cdots 0\rangle_{ACB}\langle 10\cdots 0|\right] - |0\rangle_B\langle 0| \,, \\ \Theta_B^{(2)}(\tau) := \mathrm{Tr}_{AC}\left[\mathcal{U}(\tau)\left(|00\rangle_{AB}\langle 00| \otimes \Delta\rho_C^{(\star)}[2]\right) + |0\cdots 01\rangle_{ACB}\langle 0\cdots 01|\right] - |1\rangle_B\langle 1| \,. \end{cases} \tag{74}$$

From Eqs. (28)–(29) it then follows

$$Q_C^{(\star)} = E_N\left[x_1 \chi_A^{(1)}(\tau) + x_2 \chi_A^{(2)}(\tau)\right] \,, \qquad Q_H^{(\star)} = E_1\left[x_1 \chi_B^{(1)}(\tau) + x_2 \chi_B^{(2)}(\tau)\right] \,, \tag{75}$$

where for $j = 1, 2$ we have

$$\chi_A^{(j)}(\tau) := \mathrm{Tr}\left[S_A^Z \Theta_A^{(j)}(\tau)\right] \,, \qquad \chi_B^{(j)}(\tau) := \mathrm{Tr}\left[S_V^Z \Theta_B^{(j)}(\tau)\right] \,. \tag{76}$$

These expressions can finally be simplified by invoking the symmetries (35) and (39) which applied to the present case impose the constraints

$$\chi_A^{(1)}(\tau) = -\chi_A^{(2)}(\tau) = \chi_B^{(2)}(\tau) = -\chi_B^{(1)}(\tau) \,. \tag{77}$$

Indentifying hence $f_2(\tau)$ with $-\chi_A^{(1)}(\tau)$ we can finally write

$$\begin{cases} Q_C^{(\star)} &= (x_2 - x_1)f_2(\tau)E_N \,, \\ Q_H^{(\star)} &= -(x_2 - x_1)f_2(\tau)E_1 \,, \\ W^{(\star)} &= (x_2 - x_1)f_2(\tau)(E_1 - E_N) \,, \\ \Delta\mathcal{C}^{(\star)} &= (x_2 - x_1)f_2(\tau)(\beta_2 E_N - \beta_1 E_1) \,, \end{cases} \tag{78}$$

which correspond to the low-temperature counterparts of Eq. (49).

The great advantage of Eq. (78) is that the only part of the Hilbert space that leads to useful contributions to the relevant thermodynamic quantities contains just one spin up and all spin down. This leads to an exponential speed-up in the computation of $f_2(\tau)$ which, in virtue of the ansatz (49), allows for the study of chains long up to thousands of sites, as shown in Fig. 4. A slightly more compact expression for $f_2(\tau)$ can be obtained by noticing that since the operator $\Omega := |00\rangle_{AB}\langle 00| \otimes \Delta\rho_C^{(\star)}[1] + |10\cdots 0\rangle_{ACB}\langle 10\cdots 0|$ is **bd** and contains no more than one spin up, so does its evolution under $\mathcal{U}(\tau)$. Accordingly we can write

$$\begin{aligned} {}_A\langle 0|\mathrm{Tr}_{CB}[\mathcal{U}(\tau)(\Omega)]|0\rangle_A &= \mathrm{Tr}[\mathcal{U}(\tau)(\Omega)] - {}_A\langle 1|\mathrm{Tr}_{CB}[\mathcal{U}(\tau)(\Omega)]|1\rangle_A \\ &= 1 - \langle 10\cdots 0|\mathcal{U}(\tau)(\Omega)|10\cdots 0\rangle \,, \end{aligned}$$

where used the fact that $\Omega$ (and hence $\mathcal{U}(\tau)(\Omega)$) has trace one. Expressing

$$S_Z^A = \frac{|1\rangle_A\langle 1| - |0\rangle_A\langle 0|}{2} = \frac{I_A}{2} - |0\rangle_A\langle 0| \,, \tag{79}$$

and observing that $\Theta_A^{(1)}(\tau)$ is a traceless operator we can then write

$$\begin{aligned} f_2(\tau) &= -\mathrm{Tr}\left[S_A^Z \Theta_A^{(1)}(\tau)\right] = {}_A\langle 0|\Theta_A^{(1)}(\tau)|0\rangle_A = {}_A\langle 0|\mathrm{Tr}_{CB}[\mathcal{U}(\tau)(\Omega)]|0\rangle_A \\ &= 1 - \langle 10\cdots 0|\mathcal{U}(\tau)(\Omega)|10\cdots 0\rangle \,. \end{aligned} \tag{80}$$

This can be further simplified by expanding $\Omega$ in terms of its constituents: in particular using, Eq. (71) we get

$$\begin{aligned} \langle 10\cdots 0|\mathcal{U}(\tau)(\Omega)|10\cdots 0\rangle &= \langle 10\cdots 0|\mathcal{U}(\tau)(|00\rangle_{AB}\langle 00| \otimes \Delta\rho_C^{(\star)}[1])|10\cdots 0\rangle \\ &\quad + |\langle 10\cdots 0|U(\tau)|10\cdots 0\rangle|^2 \\ &= \gamma_1\langle 10\cdots 0|\mathcal{U}(\tau)(|00\rangle_{AB}\langle 00| \otimes \varrho_C^{(\star)}[1])|10\cdots 0\rangle \\ &\quad + |\langle 10\cdots 0|U(\tau)|10\cdots 0\rangle|^2 \\ &= \gamma_1\sum_{\ell=1}^{|C|} p_\ell|\langle 10\cdots 0|U(\tau)|0\phi_\ell 0\rangle|^2 + |\langle 10\cdots 0|U(\tau)|10\cdots 0\rangle|^2 \,, \end{aligned} \tag{81}$$

where we used the fact that $U(\tau)$ leaves $\rho_C^{(\star)}[0]$ invariant, and where for $\ell \in \{1, \cdots, |C|\}$, $|\phi_\ell\rangle_C$, $p_\ell$ are respectively the eigenvectors and the associated eigenvalues of the density matrix $\varrho_C^{(\star)}[1]$. Expanding finally the norm of vector $\langle 10\cdots 0|U(\tau)$ w.r.t. to the single excitation basis of the chain $\{|10\cdots 0\rangle, |0\phi_1 0\rangle, \cdots, |0\phi_{|C|}0\rangle, |0\cdots 01\rangle\}$, i.e.,

$$\begin{aligned} 1 = \|\langle 10\cdots 0|U(\tau)\|^2 &= |\langle 10\cdots 0|U(\tau)|10\cdots 0\rangle|^2 + |\langle 10\cdots 0|U(\tau)|0\cdots 01\rangle|^2 \\ &\quad + \sum_{\ell=1}^{|C|}|\langle 10\cdots 0|U(\tau)|0\phi_\ell 0\rangle|^2 \,, \end{aligned} \tag{82}$$

we finally arrive at

$$f_2(\tau) = |\langle 10\cdots 0|U(\tau)|0\cdots 01\rangle|^2 + \sum_{\ell=1}^{|C|}(1 - \gamma_1 p_\ell)|\langle 10\cdots 0|U(\tau)|0\phi_\ell 0\rangle|^2 \,, \tag{83}$$

which explicitly fulfils the constraint (51).

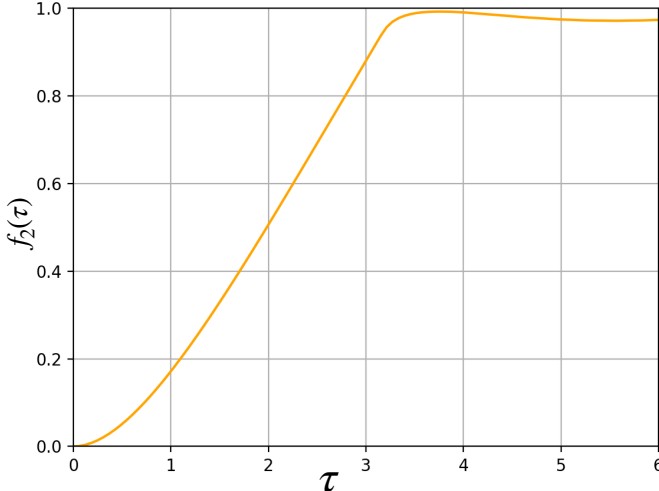

FIG. 4: Value of the function $f_2(\tau)$ of Eq. (83) computed numerically for the Hamiltonian model (33) of length $N = 1000$. Plot realized assuming $J_i = 1$, $K_i = F_i = 0$, constant over the sample, and taking $E_i$ varying linearly from $E_1 = 1$ to $E_N = 2$.

### 3. Numerical results

For large chains the problem can be approached numerically through the use of exact diagonalization techniques. In the general case, the complexity of the problem increases exponentially with the chain's size; thus, only small ones can be treated. Nonetheless we devised an algorithm that allows computing the energy exchanges at the limit cycle of the 2 stroke engine (23) for Hamiltonian chains of the form (59) – i.e., for model (33) with $K_i = F_i = 0$). All the cases analysed confirmed the behaviour reported in the ansatz (49). Examples of the result obtained are reported in Fig. 2 for $N = 8$ spins. In Fig. 4 we report instead the value of the function $f_2(\tau)$ for the case of a spin-chain of $N = 1000$ elements evaluated via Eq. (83).

### 4. remarks

We remark that the single excitation, low $T$ result of Eq. (83) holds for any temperature $T$. This per se does not mean that high $T$ effects are unimportant for calculating heat and work exchanged during the engine operation. Those are indeed important, but are fully lumped in the function $g$, rather than in the function $f$, see Eq. (49). The physical reason why the low $T$ sector of the spectrum suffices for the calculation of $f$, is that the chain is such that it allows for at most one excitation at a time to be transferred, i.e., the transfer mechanism is mediated by single excitations only. We stress that while the size of the region of validity of the low $T$ expansion may shrink with growing system size (because that would be accompanied by a shrinking of the energy gaps in the system spectrum) the low $T$ result in Eq. (83) would remain valid regardless of how small that region is (as long as it does not get exactly null).

## IV. TWO-QUBIT CHAINS WITH NO SYMMETRIES

In this section we analyse the performance of a spin chain model which does not preserve longitudinal magnetization. Under this circumstances the analysis becomes much more involved as the symmetry (39) does not apply. For the sake of simplicity we restrict to a class of $N = 2$ spin chain models described by Hamiltonians of the form

$$\begin{aligned}
H =& E_1 S_1^Z + E_2 S_2^Z + 4J_R(S_1^X S_2^X + S_1^Y S_2^Y) + 4J_I(S_1^X S_2^Y - S_1^Y S_2^X) \\
&+ 4K_R(S_1^X S_2^X - S_1^Y S_2^Y) - 4K_I(S_1^X S_2^Y + S_1^Y S_2^X) + F S_1^Z S_2^Z \,,
\end{aligned} \tag{84}$$

for which the problem can be solved analytically. As previously shown for case detailed in Sec. III B 1, by computing $U(\tau)$ one can determine the map (22) and find its fixed point $\rho_B^{(*)}$. Defining

$$\begin{cases} C_J(\tau) := \cos(\omega_J\tau) + i\sin(\omega_J\tau)\frac{E_2-E_1}{2\omega_J} \,, \\ S_J(\tau) := -\sin(\omega_J\tau)\frac{J_R-iJ_I}{2\omega_J} \,, \\ \omega_J := \frac{\sqrt{(E_2-E_1)^2+|J_R|^2+|J_I|^2}}{2} \,, \end{cases} \tag{85}$$

and

$$\begin{cases} C_K(\tau) = \cos(\omega_K\tau) + i\sin(\omega_K\tau)\frac{E_2+E_1}{2\omega_K} \,, \\ S_K(\tau) = -\sin(\omega_K\tau)\frac{K_R-iK_I}{2\omega_K} \,, \\ \omega_K = \frac{\sqrt{(E_2+E_1)^2+|K_R|^2+|K_I|^2}}{2} \,, \end{cases} \tag{86}$$

we find

$$_B\langle 1|\rho_B^{(\star)}|1\rangle_B = \frac{S(\tau_2)\left(\frac{e^{-\frac{\beta_1 E_1}{2}}}{Z_1(\beta_1)}|C_J(\tau_1)|^2+\frac{e^{\frac{\beta_1 E_1}{2}}}{Z_1(\beta_1)}|S_K(\tau_1)|^2\right)+\frac{e^{-\frac{\beta_2 E_2}{2}}}{Z_2(\beta_2)}|C_J(\tau_2)|^2+\frac{e^{\frac{\beta_2 E_2}{2}}}{Z_2(\beta_2)}|S_K(\tau_2)|^2}{1-S(\tau_1)S(\tau_2)} \,, \tag{87}$$

$$_B\langle 0|\rho_B^{(\star)}|0\rangle_B = \frac{S(\tau_2)\left(\frac{e^{\frac{\beta_1 E_1}{2}}}{Z_1(\beta_1)}|C_J(\tau_1)|^2+\frac{e^{-\frac{\beta_1 E_1}{2}}}{Z_1(\beta_1)}|S_K(\tau_1)|^2\right)+\frac{e^{\frac{\beta_2 E_2}{2}}}{Z_2(\beta_2)}|C_J(\tau_2)|^2+\frac{e^{-\frac{\beta_2 E_2}{2}}}{Z_2(\beta_2)}|S_K(\tau_2)|^2}{1-S(\tau_1)S(\tau_2)} \,, \tag{88}$$

$$_B\langle 1|\rho_B^{(\star)}|0\rangle_B = {}_B\langle 0|\rho_B^{(\star)}|1\rangle_B = 0 \,, \tag{89}$$

with $S(\tau) := |S_J(\tau)|^2 - |S_K(\tau)|^2$. Applying Eqs. (28)–(31) we arrive at

$$\begin{cases} Q_C = f_H(\tau_1,\tau_2)\frac{e^{-\frac{\beta_1 E_1}{2}}-e^{\frac{\beta_1 E_1}{2}}}{Z_1(\beta_1)} + f_C(\tau_1,\tau_2)\frac{e^{-\frac{\beta_2 E_2}{2}}-e^{\frac{\beta_2 E_2}{2}}}{Z_2(\beta_2)} \,, \\ Q_H = f_H(\tau_2,\tau_1)\frac{e^{-\frac{\beta_1 E_1}{2}}-e^{\frac{\beta_1 E_1}{2}}}{Z_1(\beta_1)} + f_C(\tau_2,\tau_1)\frac{e^{-\frac{\beta_2 E_2}{2}}-e^{\frac{\beta_2 E_2}{2}}}{Z_2(\beta_2)} \,, \\ W = -\left(f_H(\tau_1,\tau_2)+f_C(\tau_2,\tau_1)\right)\left(\frac{e^{-\frac{\beta_1 E_1}{2}}-e^{\frac{\beta_1 E_1}{2}}}{Z_1(\beta_1)}+\frac{e^{-\frac{\beta_2 E_2}{2}}-e^{\frac{\beta_2 E_2}{2}}}{Z_2(\beta_2)}\right) \,, \\ \Delta S_T = \left(\beta_1 f_C(\tau_2,\tau_1)+\beta_1 f_H(\tau_1,\tau_2)\right)\frac{e^{-\frac{\beta_1 E_1}{2}}-e^{\frac{\beta_1 E_1}{2}}}{Z_1(\beta_1)} + \left(\beta_1 f_H(\tau_2,\tau_1)+\beta_1 f_C(\tau_1,\tau_2)\right)\frac{e^{-\frac{\beta_2 E_2}{2}}-e^{\frac{\beta_2 E_2}{2}}}{Z_2(\beta_2)} \,, \end{cases}$$

where

$$f_H(\tau_1,\tau_2) := \frac{S(\tau_2)\left(|C_K(\tau_1)|^2-|S_J(\tau_1)|^2\right)^2}{1-S(\tau_1)S(\tau_2)} - |S_K(\tau_1)|^2 + |S_J(\tau_1)|^2 \,, \tag{90}$$

$$f_C(\tau_1,\tau_2) := \frac{\left(|C_K(\tau_1)|^2-|S_J(\tau_1)|^2\right)\left(|C_K(\tau_2)|^2-|S_J(\tau_2)|^2\right)}{1-S(\tau_1)S(\tau_2)} - 1 \,. \tag{91}$$

The behaviour of the machine, in this case, is much different from what we saw before; parameters that previously did not play any role in determining the operation modes now are determinant. For example, Fig. 5 shows how the free evolutions time $\tau_1$, $\tau_2$ (that previously only entered in the positive multiplying factor $f_4$ for the energy exchanges) can now modify the operation mode of the system. Despite this, certain regularities can be observed by analyzing the results numerically:

- For $E_1$ and $E_2$ having opposite signs, there are no restrictions: any thermodynamic behaviour is possible.

- For $E_1 > E_2$ and $\frac{\beta_2}{\beta_1} > \frac{E_1}{E_2}$, the system cannot operate as a refrigerator [R]: every other regime is possible.

- For $\frac{\beta_2}{\beta_1} < \frac{E_1}{E_2}$, the system can operate as a refrigerator [R] or as a heater [H].

- For $E_1 < E_2$ and $\frac{\beta_2}{\beta_1} > \frac{E_1}{E_2}$, the system behaves as an accelerator [A] or as a heater [H].

or, expressed in more compact form

$$\begin{cases} \frac{E_2}{E_1} \leq 0 & \implies [H], [R], [E], [A], \\ 0 \leq \frac{E_2}{E_1} \leq \frac{\beta_1}{\beta_2} & \implies [R], [H], \\ \frac{\beta_1}{\beta_2} \leq \frac{E_2}{E_1} \leq 1 & \implies [E], [A], [H], \\ \frac{E_2}{E_1} \geq 1 & \implies [A], [H]. \end{cases} \tag{92}$$

The same pattern was observed in Ref. [23] for a 2 spin-chain model in which the Hamiltonian coupling was replaced by the action of unital gates. Figure 6, shows the pattern of alternating operation modes in the $\tau_1$-$\tau_2$ plane for the different choices of $\frac{E_2}{E_1}$.

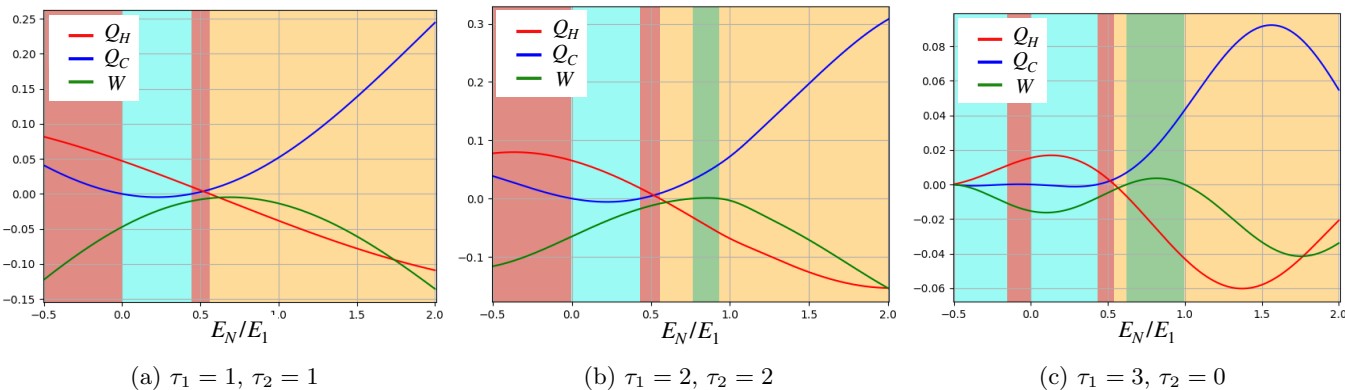

FIG. 5: Thermodynamic operation modes dependence on time for the two-qubit model with Hamiltonian (84); here $E_1 = 1$, $|J| = 1.5$, $|K| = 0.3$, $\beta_1 = 0.3$, $\beta_2 = 0.6$). The color code adopted to represent the various modes is the same we used in Fig. 2: i.e., Red [H], light blue [R], green [E], yellow [A]. $\tau_1, \tau_2$ are expressed in units of $\omega_J^{-1}$. $Q_H, Q_C, W$ are expressed in units of $E_1$.

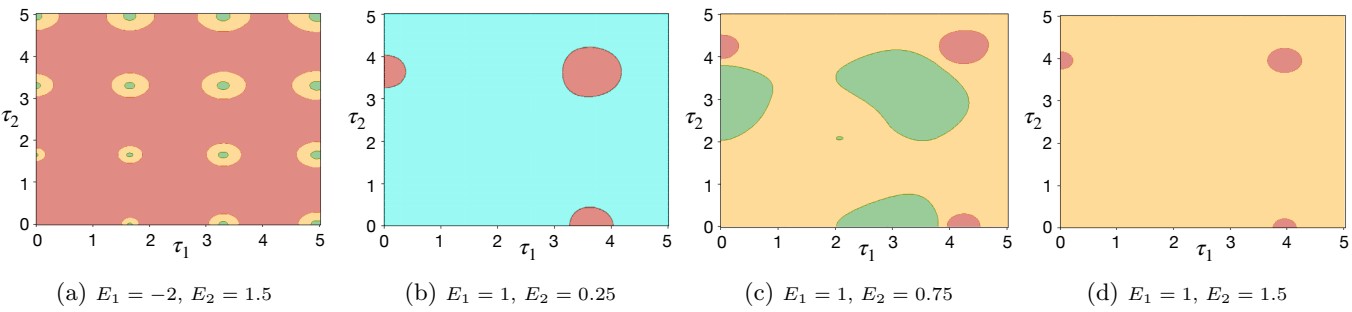

FIG. 6: Operation modes in the $\tau_1$-$\tau_2$ plane of the 2 spin chain model of (84), for various choices of $E_1$ and $E_2$ ($|J| = 1.5$, $|K| = 0.3$, $\beta_1 = 0.3$, $\beta_2 = 0.6$). As before Red areas represent [H], light blue areas represent [R], green areas represent [E], and yellow areas represent [A]. $\tau_1, \tau_2$ are expressed in units of $\omega_J^{-1}$.

## V. CONCLUSION

Our study evidences how the use of quantum channel formalism can be a valuable tool for analysing thermal engines. In particular, it shows that the presence of a global symmetry (i.e., the conservation of longitudinal magnetization), leads to a great simplification. In particular we have presented an universal law that links the thermodynamic character of the limit cycle to a finite number of parameters of the system. We also presented evidence that support a conjecture according to which for these models the fundamental thermodynamical quantities can be explicitly computed by only looking at the low temperature response of the system. Possible generalization of the present approach could be achieved by relaxing some of the technical assumptions we have adopted in the analysis, such as permitting partial thermal relaxation of the terminal elements of the network, considering more complex geometry of the spin couplings, and allowing the presence of more than two baths. We emphasize that our results are valid under the provision that the quantum channels that dictate the advancement of the chain state by one cycle, are mixing. This rather than posing a limitation on the validity of our results indeed establishes their universal character. Mixing channel are the rule rather than the exception (they form a dense sub-set of the set of all quantum channels, while non-mixing channels have zero measure), meaning that any arbitrary small random perturbation would make a generic quantum channel a mixing one. In practice it means that incurring into (or realising) a non-mixing channel is exceptionally hard. We remark that the mixing character is not associated to issues such as integrability of the chain.

### Acknowledgements

VG acknowledges financial support by MIUR (Ministero dell'Istruzione, dell'Università e della Ricerca) via project PRIN 2017 "Taming complexity via Quantum Strategies: a Hybrid Integrated Photonic approach" (QUSHIP) Id.

2017SRN- BRK, via project PRO3 Quantum Pathfinder, and PNRR MUR project PE0000023-NQSTI.

## Appendix A: Existence and uniqueness of the limit cycle for spin-chain models with magnetization preserving Hamiltonians

For spin-chain models with interactions that conserve the total longitudinal magnetization, the maps $\tilde{\Phi}_{AC}$ and $\Phi_{BC}$ introduced in Sec. II A are mixing for almost all choices of the system parameters – the only real constraint being that no exchange-interaction term is missing. While this fact can be established as a direct consequence of Refs. [38, 39, 41], for the sake of completeness we report here an explicit proof. We shall also verify that in these cases the associated fixed points states of the maps (i.e., $\rho_{CB}^{(\star)}$ and $\tilde{\rho}_{AC}^{(\star)}$) commute with their corresponding longitudinal magnetization operators (i.e., $S_{CB}^Z$ and $S_{AC}^Z$ respectively).

The material is organized as follows: in **Observation 1** we show that if one of the channels $\Phi_{BC}$ and $\tilde{\Phi}_{AC}$ is mixing also the other must share the same property (this is true in general, not just for the spin-chain models we consider here) next, in **Observation 2** we prove Eq. (34), i.e., that if the map $\Phi_{BC}$ (resp. $\tilde{\Phi}_{AC}$) of the spin-chain model of Sec. II A is mixing then its fixed point state $\rho_{CB}^{(\star)}$ ($\tilde{\rho}_{AC}^{(\star)}$) must commute with the longitudinal magnetization operator $S_{CB}^Z$ ($S_{AC}^Z$); in **Observation 3** we show that to prove that $\Phi_{BC}$ is mixing it is sufficient to verify that such property holds for the case in which $T_1 = T_2 = 0$; finally in **Observation 4** we prove that indeed at zero temperature, the channel $\Phi_{BC}$ of the spin-chain model Sec. II A is mixing under rather general assumption on the system Hamiltonian.

### Observation 1

To begin with let's first observe that by construction, irrespectively from the specific model we choose, the channel $\Phi_{BC}$ is mixing if and only if $\tilde{\Phi}_{AC}$ share the same property. Indeed given $\rho_{ACB}^{(m)}$ and $\tilde{\rho}_{ACB}^{(m)}$ respectively the states of the chain at the beginning of stroke 1 and 3 of the $m$-th cycle, it holds

$$\tilde{\rho}_{ACB}^{(m)} = \mathcal{U}_1 \circ \mathcal{T}_1(\rho_{ACB}^{(m)}) = \mathcal{U}_1(\rho_A(\beta_1) \otimes \rho_{CB}^{(m)}) \,, \tag{A1}$$

$$\rho_{ACB}^{(m+1)} = \mathcal{U}_2 \circ \mathcal{T}_2(\tilde{\rho}_{ACB}^{(m)}) = \mathcal{U}_2(\tilde{\rho}_{AC}^{(m)} \otimes \rho_B(\beta_2)) \,, \tag{A2}$$

with $\mathcal{T}_{1,2}$ the local thermalization maps of stroke 1 and 3, and with $\mathcal{U}_{1,2}$ the unitary evolutions of strokes 2 and 4. Accordingly invoking (20) and (21) we can write

$$\begin{aligned}
\Phi_{CB}^m(\rho_{CB}) &= \rho_{CB}^{(m)} = \text{Tr}_A[\mathcal{T}_1(\rho_{ACB}^{(m)})] = \text{Tr}_A[\mathcal{T}_1 \circ \mathcal{U}_2(\tilde{\rho}_{AC}^{(m-1)} \otimes \rho_B(\beta_2))] \\
&= \text{Tr}_A[\mathcal{T}_1 \circ \mathcal{U}_2(\tilde{\Phi}_{AC}^{m-2}(\tilde{\rho}_{AC}^{(1)}) \otimes \rho_B(\beta_2))] \,,
\end{aligned} \tag{A3}$$

$$\begin{aligned}
\tilde{\Phi}_{AC}^m(\tilde{\rho}_{AC}) &= \tilde{\rho}_{AC}^{(m)} = \text{Tr}_B[\mathcal{T}_2(\tilde{\rho}_{ACB}^{(m)})] = \text{Tr}_A[\mathcal{T}_2 \circ \mathcal{U}_1(\rho_A(\beta_1) \otimes \rho_{CB}^{(m)})] \\
&= \text{Tr}_B[\mathcal{T}_2 \circ \mathcal{U}_1(\rho_A(\beta_1) \otimes \Phi_{CB}^{m-1}(\rho_{CB}^{(1)}))] \,,
\end{aligned} \tag{A4}$$

which link the asymptotic behaviours of the two channels relating their fixed points as in Eq. (26). Thanks to this observation our task can hence be reduced to the study of the mixing property of $\Phi_{CB}$ only.

### Observation 2

Here we show that in the case of the spin-chain models of Sec. II A that conserves the total longitudinal magnetization, if $\Phi_{CB}$ ($\tilde{\Phi}_{AC}$) is mixing, then its unique fixed point state $\rho_{CB}^{(\star)}$ (resp. $\tilde{\rho}_{AC}^{(\star)}$) must commute with the longitudinal magnetization operator $S_{CB}^Z$ (resp. $S_{AC}^Z$).

To show this fact given $\mathcal{H}_D$ the Hilbert space associated with a portion $D$ of the spin-chain, consider its decomposition as a direct sum in terms of the eigenspaces $\mathcal{H}_{D,n}$ of $S_D^Z$, i.e., $\mathcal{H}_D = \oplus_{n=0}^{|D|} \mathcal{H}_{D,n}$ where $\mathcal{H}_{D,n}$ represents the subspace of the system formed by the vectors in which we have exactly $n$ particle that have spin up along the $z$-th direction and the remaining $|D| - n$ ones which are pointing spin down ($|D|$ representing here the total number of spin in $D$). Now introducing $\Pi_D(n)$ the projector on $\mathcal{H}_{D,n}$, given a generic operator $\Theta_D$ acting on $\mathcal{H}_D$, let us decompose it as

$$\Theta_D = \Theta_D^{(\mathbf{bd})} + \Theta_D^{(\mathbf{off})} \,, \qquad \begin{cases} \Theta_D^{(\mathbf{bd})} := \sum_{n=0}^{|D|} \Pi_D(n) \Theta_D \Pi_D(n) \,, \\ \\ \Theta_D^{(\mathbf{off})} := \sum_{n \neq n'} \Pi_D(n) \Theta_D \Pi_D(n') \,, \end{cases} \tag{A5}$$

where $\Theta_D^{(\mathbf{bd})}$ represents the block diagonal (**bd**) part of $D$ which, for all $n$, sends $\mathcal{H}_{D,n}$ into $\mathcal{H}_{D,n}$, and with $\Theta_D^{(\mathbf{off})}$ the trace-zero, (off-diagonal) contribution which instead induces transitions among the various subspace $\mathcal{H}_{D,n}$. The following properties are easy to check:

*i)* an operator $\Theta_D$ can commute with $S_D^Z$ if and only if it is **bd** (i.e., iff its off-diagonal part $\Theta_D^{(\mathbf{off})}$ is null);

*ii)* any unitary evolution $\mathcal{U}$ which preserves the longitudinal magnetization of $D$ will send **bd** operators of $D$ into **bd** operators.

Furthermore, since the subspaces $\mathcal{H}_{D,n}$ of a composite system $D = D_1 D_2$ of two distinct portions of chain decompose as direct sums $\mathcal{H}_{D,n} = \oplus_{n_1=0}^n \mathcal{H}_{D_1,n_1} \otimes \mathcal{H}_{D_2,n-n_1}$ with $\mathcal{H}_{D_j,n_j}$ being the eigenspace of $D_j$ with $n_j$ spins-up, we also have that

*iii)* the product states $\rho_{D_1} \otimes \rho_{D_2}$ of local density matrices $\rho_{D_1}$, $\rho_{D_2}$ which are locally **bd**, is globally **bd**;

*iv)* the reduced density matrices of $\rho_{D_1} = \text{Tr}_{D_2}[\rho_D]$, $\rho_{D_2} = \text{Tr}_{D_1}[\rho_D]$ of a state $\rho_D$ which is globally **bd**, are locally **bd**.

Notice that from *iii)* and *iv)* it follows that since that, for all temperatures, the Gibbs states $\rho_A(\beta_1)$ and $\rho_B(\beta_2)$ are locally **bd**, we have that the thermal maps $\mathcal{T}_1$ and $\mathcal{T}_2$ which enter in the definitions (18) and (21) are **bd** preserving transformations. Since due *ii)* the same property holds for $\mathcal{U}_1$ and $\mathcal{U}_2$ due to, we can conclude that $\Phi_{CB}$ and $\tilde{\Phi}_{AC}$ also map **bd** states into **bd** states. This means that the input states which are **bd** will maintain such property even after repeated applications of $\Phi_{CB}$ ($\tilde{\Phi}_{AC}$): considering that in case the is mixing such states will be driven toward the unique fix point $\rho_{CB}^{(\star)}$ (resp. $\tilde{\rho}_{AC}^{(\star)}$) we can conclude that the latter must be **bd** as well or, in view of property *i)*, that this state commutes with the longitudinal magnetization operator of the chain $S_{CB}^Z$ (resp. $S_{AC}^Z$).

### Observation 3

A further important simplification arises by noticing that from Eq. (18) it follows that such channel can be decomposed as the convex sum of a collection of independent CPTP terms

$$\Phi_{CB}(\cdots) = \sum_{i,j'} \frac{e^{-(\beta_1 \epsilon_i^A + \beta_2 \epsilon_{j'}^B)}}{Z_A(\beta_1) Z_B(\beta_2)} \Phi_{CB}^{(i,j')}(\cdots) , \tag{A6}$$

where introducing $\mathcal{T}_2^{(j')} := \text{Tr}_B[\cdots] \otimes |j'\rangle_B \langle j'|$ the LCPT transformation that replace the $B$ section of the chain with the $j'$-th energy eigenstate of $H_B$, we have

$$\Phi_{CB}^{(i,j')}(\cdots) := \text{Tr}_A \left[ \mathcal{U}_2 \circ \mathcal{T}_2^{(j')} \circ \mathcal{U}_1 (|i\rangle_A \langle i| \otimes \cdots) \right] = \sum_{i',j} S_{\alpha,\alpha'}(\cdots) S_{\alpha,\alpha'}^\dagger , \tag{A7}$$

with $\alpha = (i,j)$ a joint index and

$$S_{\alpha,\alpha'} := {}_A\langle j|U(\tau_1)|i\rangle_B \langle i'|U(\tau_2)|j'\rangle_A , \tag{A8}$$

the associated Kraus set. In particular, identifying with $\epsilon_0^A$ and $\epsilon_0^B$ with the ground states of $H_A$ and $H_B$, it follows that the term $\Phi_{CB}^{(0,0)}$ of (A7) represents the map one would obtain by setting the system temperatures equal to zero, $T_1 = T_2 = 0$ (its weight being the largest in the decomposition). Accordingly invoking the fact that mixing channels are stable under randomization [39] we can claim that if $\Phi_{CB}^{(0,0)}$ is mixing, also $\Phi_{CB}$ (and thus $\tilde{\Phi}_{AC}$) will be mixing for any other choice of the temperatures.

### Observation 4

Here we prove that a part from very special cases where the system Hamiltonian $H$ admits at least an eigenvector of the factorized form

$$|E\rangle_{ACB} = |0\rangle_A \otimes |\phi\rangle_C \otimes |0\rangle_B , \tag{A9}$$

the zero-temperature channel $\Phi_{CB}^{(0,0)}$ introduced in the previous section is indeed mixing. As shown in Ref. [41] the condition (A9) is indeed rare as it cannot occur as long as all the nearest-neighbor interactions of the model contain exchange parts (i.e., if $|J_i| + |K_i| \neq 0$ for all $i$).

By direct computation it is easy to verify that $\Phi_{CB}^{(0,0)}$ admits as fixed point the state $|0\cdots 0\rangle_{CB}$ in which all the spins are pointing down along the $z$-th direction. Therefore if $\Phi_{CB}^{(0,0)}$ is mixing we must have that

$$\lim_{m\to\infty} (\Phi_{CB}^{(0,0)})^m(\rho_{CB}) = |0\cdots 0\rangle_{CB}\langle 0\cdots 0| \ , \tag{A10}$$

for all states $\rho_{CB}$. Notice that we can restrict the analysis to the cases in which $\rho_{CB}$ are **bd** states: indeed if (A10) applies to all such configurations then, from Eq. (A5) it will follows that in the limit $m \to \infty$ a generic (not-necessarily **bd**) state will be driven into a final configuration that has $|0\cdots 0\rangle_{CB}\langle 0\cdots 0|$ as the **bd** component, but since $|0\cdots 0\rangle_{CB}\langle 0\cdots 0|$ is pure, this is indeed the only state that fulfil such property. Now given $\rho_{CB}$ **bd** a state, using the fact that $\Phi_{CB}$ is a **bd** preserving channel (see Observation 2) it follows that its evolved counterpart under $m$ iterate actions of $\Phi_{CB}^{(0,0)}$ can be expressed as

$$\rho_{CB}^{(m)} \ := \ (\Phi_{CB}^{(0,0)})^m(\rho_{CB}) = \sum_{n=0}^{|CB|} \Pi_{CB}(n)\rho_{CB}^{(m)}\Pi_{CB}(n) = \sum_{n=0}^{|CB|} q_n^{(m)}\rho_{CB}^{(m)}(n) \ , \tag{A11}$$

with $q_n^{(m)} := \mathrm{Tr}_{CB}\left[\Pi_{CB}(n)\rho_{CB}^{(m)}\right]$ the probability that the system contains exactly $n$ spins up in $CB$ and $\rho_{CB}^{(m)}(n) := \Pi_{CB}(n)\rho_{CB}^{(m)}\Pi_{CB}(n)/q_n^{(m)}$ (notice that for $m = 0$ the above equation simply represents the **bd** decomposition (A5) of the input state). By construction the quantity

$$P_n^{(m)}(\rho_{CB}) := \sum_{n'=n}^{|CB|} q_{n'}^{(m)} = \sum_{n'=n}^{|CB|} \mathrm{Tr}_{CB}\left[\Pi_{CB}(n')\rho_{CB}^{(m)}\right] \ , \tag{A12}$$

measures the fraction of $\rho_{CB}$ which contains at least $n$ spins up after $m$ iterated applications of the channel $\Phi_{CB}^{(0,0)}$: proving (A10) accounts to show that, for all $n \geq 1$, $P_n^{(m)}$ converges to zero as $m$ goes to infinity for all $\rho_{CB}$. Observe that for each fixed $m$ and $\rho_{CB}$, $P_n^{(m)}(\rho_{CB})$ is not increasing w.r.t. $n$, i.e.

$$P_{n+1}^{(m)}(\rho_{CB}) \leq P_n^{(m)}(\rho_{CB}) \leq P_0^{(m)}(\rho_{CB}) = 1 \ . \tag{A13}$$

Furthermore since the zero-temperture transformation $\Phi_{CB}^{(0,0)}$ cannot increase the longitudinal magnetization of the input states (indeed the unitary $\mathcal{U}_1$ and $\mathcal{U}_2$ preserve the magnetization, while $\mathcal{T}_1$ and $\mathcal{T}_2$ replace part of their input states with terms that contains no spin-up components), we have that for all $m$, $\Delta m$ non negative integers, one has

$$\mathrm{Tr}[\Pi_{CB}(n')(\Phi_{CB}^{(0,0)})^{\Delta m}(\rho_{CB}^{(m)}(n''))] = 0 \ , \qquad \forall n' > n'' \ , \tag{A14}$$

which in particular implies that for each given $n \geq 1$ and $\rho_{CB}$, $P_n^{(m)}(\rho_{CB})$ is also a non decreasing function of $m$, i.e.,

$$P_n^{(m+1)}(\rho_{CB}) \leq P_n^{(m)}(\rho_{CB}) \ . \tag{A15}$$

Following Ref. [41] given any positive integer $\Delta m$ and $n \leq |CB| - 1$ we can then derive the inequalities

$$
\begin{aligned}
P_n^{(m+\Delta m)}(\rho_{CB}) &= \sum_{n' \geq n} \mathrm{Tr}_{CB}\left[\Pi_{CB}(n')(\Phi_{CB}^{(0,0)})^{\Delta m}(\rho_{CB}^{(m)})\right] \\
&= \sum_{n' \geq n} \sum_{n'' \geq n} q_{n''}^{(m)} \mathrm{Tr}_{CB}\left[\Pi_{CB}(n')(\Phi_{CB}^{(0,0)})^{\Delta m}\left(\rho_{CB}^{(m)}(n'')\right)\right] \\
&= \sum_{n' \geq n} \sum_{n'' \geq n+1} q_{n''}^{(m)} \mathrm{Tr}_{CB}\left[\Pi_{CB}(n')(\Phi_{CB}^{(0,0)})^{\Delta m}\left(\rho_{CB}^{(m)}(n'')\right)\right] \\
&\quad + q_n^{(m)} \mathrm{Tr}_{CB}\left[\Pi_{CB}(n)(\Phi_{CB}^{(0,0)})^{\Delta m}\left(\rho_{CB}^{(m)}(n)\right)\right] \\
&\leq \sum_{n' \geq n} \sum_{n'' \geq n+1} q_{n''}^{(m)} \mathrm{Tr}_{CB}\left[\Pi_{CB}(n')\rho_{CB}^{(m)}(n'')\right] \\
&\quad + q_n^{(m)} \mathrm{Tr}_{CB}\left[\Pi_{CB}(n)(\Phi_{CB}^{(0,0)})^{\Delta m}\left(\rho_{CB}^{(m)}(n)\right)\right] \\
&= \sum_{n' \geq n+1} \sum_{n'' \geq n+1} q_{n''}^{(m)} \mathrm{Tr}_{CB}\left[\Pi_{CB}(n')\rho_{CB}^{(m)}(n'')\right] \\
&\quad + q_n^{(m)} \mathrm{Tr}_{CB}\left[\Pi_{CB}(n)(\Phi_{CB}^{(0,0)})^{\Delta m}\left(\rho_{CB}^{(m)}(n)\right)\right] \\
&= \sum_{n' \geq n+1} \mathrm{Tr}_{CB}\left[\Pi_{CB}(n')(\Phi_{CB}^{(0,0)})^{m}(\rho_{CB})\right] \\
&\quad + q_n^{(m)} \mathrm{Tr}_{CB}\left[\Pi_{CB}(n)(\Phi_{CB}^{(0,0)})^{\Delta m}\left(\rho_{CB}^{(m)}(n)\right)\right] \\
&\leq P_{n+1}^{(m)}(\rho_{CB}) + Q_n^{(m \mapsto m+\Delta m)} \,,
\end{aligned} \tag{A16}
$$

where

$$
Q_n^{(m \mapsto m+\Delta m)} := \max_{|\psi_n\rangle_{CB} \in \mathcal{H}_{CB,n}} \mathrm{Tr}_{CB}\left[\Pi_{CB}(n)\Phi_{CB}^{\Delta m}(|\psi_n\rangle_{CB}\langle\psi_n|)\right] \,, \tag{A17}
$$

is the maximum population of a generic state $|\psi_n\rangle_{CB} \in \mathcal{H}_{CB,n}$ which after $m$ iteration that also remain in $\mathcal{H}_{CB,n}$ after $\Delta$ extra steps. For the special case in which $n = |CB|$, the same analysis holds, obtaining

$$
P_{|CB|}^{(m+\Delta m)}(\rho_{CB}) \leq +Q_{|CB|}^{(m \mapsto m+\Delta m)} \,. \tag{A18}
$$

From here the analysis exactly mimics the one of Ref. [41]. We can in fact express $Q_n^{(m \mapsto m+\Delta m)}$ as

$$
\begin{aligned}
Q_n^{(m \mapsto m+\Delta m)} &= \max_{|\psi_n\rangle_{CB} \in \mathcal{H}_{CB,n}} ||(P_2 P_1)^{\Delta m} |0\rangle_A \otimes |\psi_n\rangle_{CB}||^2 \\
&= \max_{|\psi_n\rangle_{CB} \in \mathcal{H}_{CB,n}} ||(P_{2,n} P_{1,n})^{\Delta m} |0\rangle_A \otimes |\psi_n\rangle_{CB}||^2 \,,
\end{aligned} \tag{A19}
$$

where $P_1 := |0\rangle_B\langle 0|U(\tau_1)$, $P_2 := |0\rangle_A\langle 0|U(\tau_2)$, $P_{i,n} := \Pi_{ACB}(n)P_i\Pi_{ACB}(n)$ their restrictions to the subspaces $\mathcal{H}_{ACB,n}$. Now from the spectral properties of the projector operators it follows that un less the Hamiltonian $H$ admits at least an eigenvector of the form (A9) we get

$$
\lim_{\Delta m \to \infty} Q_n^{(m \mapsto m+\Delta m)} = 0 \,, \qquad \forall n > 1, \forall m \,. \tag{A20}
$$

Replacing this into Eq. (A22) and (A16) respectively we finally get

$$
\lim_{m \to \infty} P_{|CB|}^{(m)}(\rho_{CB}) = 0 \,, \tag{A21}
$$

and

$$
\lim_{m \to \infty} P_n^{(m)}(\rho_{CB}) = \lim_{m \to \infty} P_{n+1}^{(m)}(\rho_{CB}) = \cdots \lim_{m \to \infty} P_{|CB|}^{(m)}(\rho_{CB}) = 0 \,. \tag{A22}
$$

---

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
