# Peer review of "Spin-chain based quantum thermal machines"

_SciPost Physics, doi:SciPost Phys. 18, 160 (2025)_

## Round 1 · Referee Report · Anonymous (Referee 1) · 2024-4-1

Report

The authors have considered and addressed satisfactorily all the points raised in the referee report.

The only thing is that I could not see footnote [42] in the pdf file I had access to, however I agree with what the authors have written in the reply to my comments/questions. So I think that the paper is ready for publication.
  • validity: -
  • significance: -
  • originality: -
  • clarity: -
  • formatting: -
  • grammar: -

Author:  Vittorio Giovannetti  on 2025-04-15  [id 5373]

(in reply to Report 1 on 2024-04-01)

We thank the referee for his/her positive comment.

---

## Round 1 · Referee Report · Anonymous (Referee 2) · 2024-4-3

Report

I am happy with the changes made by the authors. I recommend publication in SciPost physics if the changes below are addressed.

Requested changes

  1. It occurred to me on a second reading that, although the validity of Eq 49 for larger spin chains is asserted, no explicit numerical evidence for this is given. Could this be included? (For example, one could plot Qh and Qc as a function of tau for different temperatures and show that the curves are universal when divided by g (Eq 50).)
  2. It may be good to state the values of tau used for Fig 2, for reproducibility.
  3. There are some other works on spin chain engines that the authors may wish to reference, for example Phys. Rev. Research 2, 043247 (2020), Phys. Rev. E 102, 012138 (2020), Phys. Rev. Research 2, 023145 (2020), Phys. Rev. B 109, 024310 (2024).

  • validity: good
  • significance: good
  • originality: good
  • clarity: ok
  • formatting: good
  • grammar: good

Author:  Vittorio Giovannetti  on 2025-04-15  [id 5372]

(in reply to Report 2 on 2024-04-03)
Category:
answer to question

1. Following the referee’s suggestion we added a plot of $Q_C/g E_N$ for a 4 spin chain. The plots shows that clearly there is no dependence on $\beta_1$. The same identical plot results when plotting vs $\beta_2$ at fixed $\beta_1$. Since Q_C/E_N=-Q_H/E_1, the same holds for Q_H/g E_1 as well, so there is no need to add these extra plots.

2. We added the value of $\tau$ used for figure 2

3. We thank the referee for pointing those references, which now have been added to the manuscript.

---

## Round 1 · Author Response

Dear Editor,
We would like to thank you and the referees for reviewing our work titled:
Spin-chain based quantum thermal machines
The referees raised a number of pertinent questions which we all addressed below in our replies, and implemented in the manuscript. Amendments include added text, modified text, improved figures, references added, minor typos corrected. Our replies below are in blue and the text changes are highlighted in red for your convenience.
We think that this new version now meets the publication criteria of SciPost and hope to see our work soon disseminated through it.
Sincerely,
Vittorio Giovannetti, Michele Campisi, Edoardo Maria Centamori

---

## Round 1 · List of Changes

We have modified the paper following the referee suggestions. In particular 1) a sentence has been added in the introductory section to specify what type of external driving we use in the protocol 2) The introductory part of Sec. II A has been re-edited to improve the presentation of how the LCPTP formalism is used in the analysis. 3) A sentence has been modified in Sec. II B to improve the presentation of the mixing property of LCPTP maps. 4) A new section has been added at the end of Sec. III B to better explain the low-temperature analysis of the model. 5) A new sentence has been added at the end of the conclusions. 6) The quality of the figures has been improved following the suggestions of the referee. 7) Minor editing and typos corrections.

---

## Round 2 · Author Response

Dear Editor, We would like to thank you and the referees for reviewing our work titled: Spin-chain based quantum thermal machines Both referee recommend publication, and referee 1 had a few final remarks which we addressed in our replies below and in the text. We think that this new version now meets the publication criteria of SciPost and hope to see our work soon disseminated through it. Sincerely, Vittorio Giovannetti and Michele Campisi

Anonymous Report 2 on 2024-4-3 (Invited Report) Report I am happy with the changes made by the authors. I recommend publication in SciPost physics if the changes below are addressed. Requested changes 1. It occurred to me on a second reading that, although the validity of Eq 49 for larger spin chains is asserted, no explicit numerical evidence for this is given. Could this be included? (For example, one could plot Qh and Qc as a function of tau for different temperatures and show that the curves are universal when divided by g (Eq 50).) 2. It may be good to state the values of tau used for Fig 2, for reproducibility. 3. There are some other works on spin chain engines that the authors may wish to reference, for example Phys. Rev. Research 2, 043247 (2020), Phys. Rev. E 102, 012138 (2020), Phys. Rev. Research 2, 023145 (2020), Phys. Rev. B 109, 024310 (2024). REPLY: 1. Following the referee’s suggestion we added a plot of $Q_C/g E_N$ for a 4 spin chain. The plots shows that clearly there is no dependence on $\beta_1$. The same identical plot results when plotting vs $\beta_2$ at fixed $\beta_1$. Since Q_C/E_N=-Q_H/E_1, the same holds for Q_H/g E_1 as well, so there is no need to add these extra plots. 2. We added the value of $\tau$ used for figure 2 3. We thank the referee for pointing those references, which now have been added to the manuscript.

---

## Round 2 · List of Changes

1. A new figure has been introduced (Fig. 3) which presents the plot of $Q_C/g E_N$ for a 4 spin chain.
2. We added the value of $\tau$ used for figure 2
3. The references Phys. Rev. Research 2, 043247 (2020), Phys. Rev. E 102, 012138 (2020), Phys. Rev. Research 2, 023145 (2020), Phys. Rev. B 109, 024310 (2024) have been added to the manuscript.

---

## Editorial Decision

published